# Identification of ubiquitin Ser57 kinases regulating the oxidative stress response in yeast

Nathaniel L Hepowit[1], Kevin N Pereira[1], Jessica M Tumolo[1], Walter J Chazin[2], Jason A MacGurn[1]*

[1]Department of Cell and Developmental Biology, Vanderbilt University, Nashville, United States; [2]Department of Biochemistry, Vanderbilt University, Nashville, United States

**Abstract** Ubiquitination regulates many different cellular processes, including protein quality control, membrane trafficking, and stress responses. The diversity of ubiquitin functions in the cell is partly due to its ability to form chains with distinct linkages that can alter the fate of substrate proteins in unique ways. The complexity of the ubiquitin code is further enhanced by post-translational modifications on ubiquitin itself, the biological functions of which are not well understood. Here, we present genetic and biochemical evidence that serine 57 (Ser57) phosphorylation of ubiquitin functions in stress responses in *Saccharomyces cerevisiae*, including the oxidative stress response. We also identify and characterize the first known Ser57 ubiquitin kinases in yeast and human cells, and we report that two Ser57 ubiquitin kinases regulate the oxidative stress response in yeast. These studies implicate ubiquitin phosphorylation at the Ser57 position as an important modifier of ubiquitin function, particularly in response to proteotoxic stress.

*For correspondence:
jason.a.macgurn@vanderbilt.edu

Competing interests: The authors declare that no competing interests exist.

## Introduction

Ubiquitin is a post-translational modifier that regulates diverse cellular processes in eukaryotic cells. The broad utility of ubiquitin as a regulatory modification is due to the high degree of complexity associated with ubiquitin polymers, which are added to substrate proteins by the activity of ubiquitin conjugation machinery (E1-E2-E3 cascades) and removed from substrates by deubiquitylases (DUBs). Ubiquitin can be conjugated recursively at any of seven internal lysines or the N-terminus to generate polymers with distinct topological features (*Herhaus and Dikic, 2015*; *Yau and Rape, 2016*; *Swatek and Komander, 2016*). Complexity is further enhanced by the formation of mixed and branched polymers (*Ohtake et al., 2018*; *Meyer and Rape, 2014*; *Swatek et al., 2019*) and by post-translational modifications that can occur on ubiquitin itself (*Herhaus and Dikic, 2015*). For example, PINK1-mediated phosphorylation of ubiquitin at the Ser65 position plays an important role in mitophagy by regulating parkin-mediated ubiquitination of mitochondrial membrane proteins (*Wauer et al., 2015*; *Ordureau et al., 2015*; *Koyano et al., 2014*; *Kazlauskaite et al., 2014*; *Kane et al., 2014*; *Ordureau et al., 2014*). Phosphorylation of ubiquitin at the Ser57 has also been reported (*Peng et al., 2003*; *Swaney et al., 2015*; *Lee et al., 2017*), but the kinases that produce this modification and its regulatory significance remain unknown.

Many proteotoxic stresses activate ubiquitin networks to promote protein quality control and protect the cell from damage associated with systemic protein misfolding. Oxidative stress is highly damaging to the cell, triggering deployment and re-distribution of existing ubiquitin pools and induction of ubiquitin biosynthesis to promote survival by activating a repertoire of ubiquitin-mediated responses (*Cheng et al., 1994*). During oxidative stress, many proteins become damaged and

misfolded, resulting in a global increase in K48-linked ubiquitin conjugation that targets substrates for clearance by proteasome-mediated degradation (*Finley, 2009*; *Shang and Taylor, 2011*). Oxidative stress also triggers translation arrest (*Grant, 2011*), resulting in K63-linked polyubiquitylation on ribosomes to stabilize the 80S complex and the formation of polysomes (*Silva et al., 2015*). Furthermore, oxidative damage of DNA activates signaling and repair processes that are tightly regulated by K63 ubiquitylation and deubiquitylation activities (*Demple and Harrison, 1994*; *Bergink and Jentsch, 2009*; *Ng et al., 2016*; *Croteau and Bohr, 1997*; *Thorslund et al., 2015*). Thus, ubiquitin networks regulate many cellular processes critical for survival during conditions of oxidative stress.

Although ubiquitin networks play a critical role in the eukaryotic cellular response to proteotoxic stress, precisely how these networks are tuned to enhance protein quality control and other protective functions remain unclear. Given that Ser65 phosphorylation of ubiquitin regulates the clearance of damaged mitochondria (*Wauer et al., 2015*; *Ordureau et al., 2015*; *Koyano et al., 2014*; *Kazlauskaite et al., 2014*; *Kane et al., 2014*; *Ordureau et al., 2014*), we hypothesized that phosphorylation at other positions may regulate ubiquitin function, particularly in conditions that promote protein damage and misfolding. Since it is the most abundant phosphorylated form (*Swaney et al., 2015*), we examined the biological functions of Ser57 phosphorylated ubiquitin in yeast, aiming to identify and characterize the molecular events and signaling processes that regulate its production.

## Results

To probe potential biological functions of Ser57 ubiquitin phosphorylation in yeast, we generated yeast strains expressing exclusively wildtype, Ser57Ala (phosphorylation resistant, or S57A) or Ser57-Asp (phosphomimetic, or S57D) ubiquitin. It is important to emphasize that such complete ubiquitin replacement may exaggerate effects associated with physiological ubiquitin phosphorylation, which occurs at very low stoichiometry (*Swaney et al., 2015*; *Lee et al., 2017*) and probably in a highly localized manner (as exemplified by PINK1-mediated Ser65 phosphorylation of ubiquitin on damaged segments of mitochondrial membrane [*Pickrell and Youle, 2015*]). With these limitations in mind, we examined the growth of these yeast strains in the context of various stressors, including heat stress, DNA damage and replication stress (hydroxyurea and arsenate), and protein misfolding stress (canavanine and thialysine, which are toxic analogs of arginine and lysine, respectively). We found that expression of S57A or S57D ubiquitin did not affect growth at ambient temperature (26° C) (consistent with previous reports [*Peng et al., 2003*; *Lee et al., 2017*; *Sloper-Mould et al., 2001*]) or sensitivity to arsenate (*Figure 1A*). However, expression of S57D ubiquitin conferred sensitivity to hydroxyurea and resistance to canavanine (which was reported previously [*Lee et al., 2017*]) and thialysine (*Figure 1A*). We also noticed that expression of S57D ubiquitin enhanced both long-term and acute tolerance of thermal stress (*Figure 1B–D*) while yeast expressing S57A ubiquitin exhibited thermal sensitivity (*Figure 1E* and *Figure 1—figure supplement 1*). These findings indicate that Ser57 phosphorylation of ubiquitin promotes cellular tolerance of various proteotoxic stressors.

Next, we analyzed the role of Ser57 phosphorylation in the oxidative stress response. Wildtype yeast cells arrest growth in response to oxidative stress and activate responses that help cells cope with the proteotoxic and DNA damaging effects of oxidation (*Silva et al., 2015*; *Shapira et al., 2004*; *Petti et al., 2011*; *Martindale and Holbrook, 2002*). Interestingly, while yeast cells expressing wildtype or S57D ubiquitin arrested growth in response to moderate oxidative stress (>1 mM $H_2O_2$), cells expressing S57A ubiquitin were deficient in this response and only arrested growth in response to more severe oxidative stress (>2 mM $H_2O_2$) (*Figure 2A*). The failed growth arrest observed for cells expressing S57A ubiquitin correlated with decreased viability (*Figure 2B*). Since oxidative stress induces the production of both K48- and K63-linked ubiquitin conjugates (*Shang and Taylor, 2011*; *Silva et al., 2015*; *Sun et al., 2009*; *Tsirigotis et al., 2001*) we tested if the expression of S57A and S57D ubiquitin alters ubiquitin conjugation patterns in response to oxidative stress. We found that 30 min exposure to $H_2O_2$ (1.0 mM) resulted in an increased abundance of total ubiquitin conjugates (*Figure 2C*) as well as K48-linked polymers (*Figure 2C–D*), consistent with a previous report (*Silva et al., 2015*). Compared to cells expressing wildtype ubiquitin, we found that cells expressing S57D ubiquitin exhibited increased abundance of K48-linked polymers and decreased abundance of K63-linked polymers (*Figure 2C–E*). By contrast, cells expressing S57A ubiquitin did not exhibit a significant increase in the production of K48-linked polyubiquitin chains in

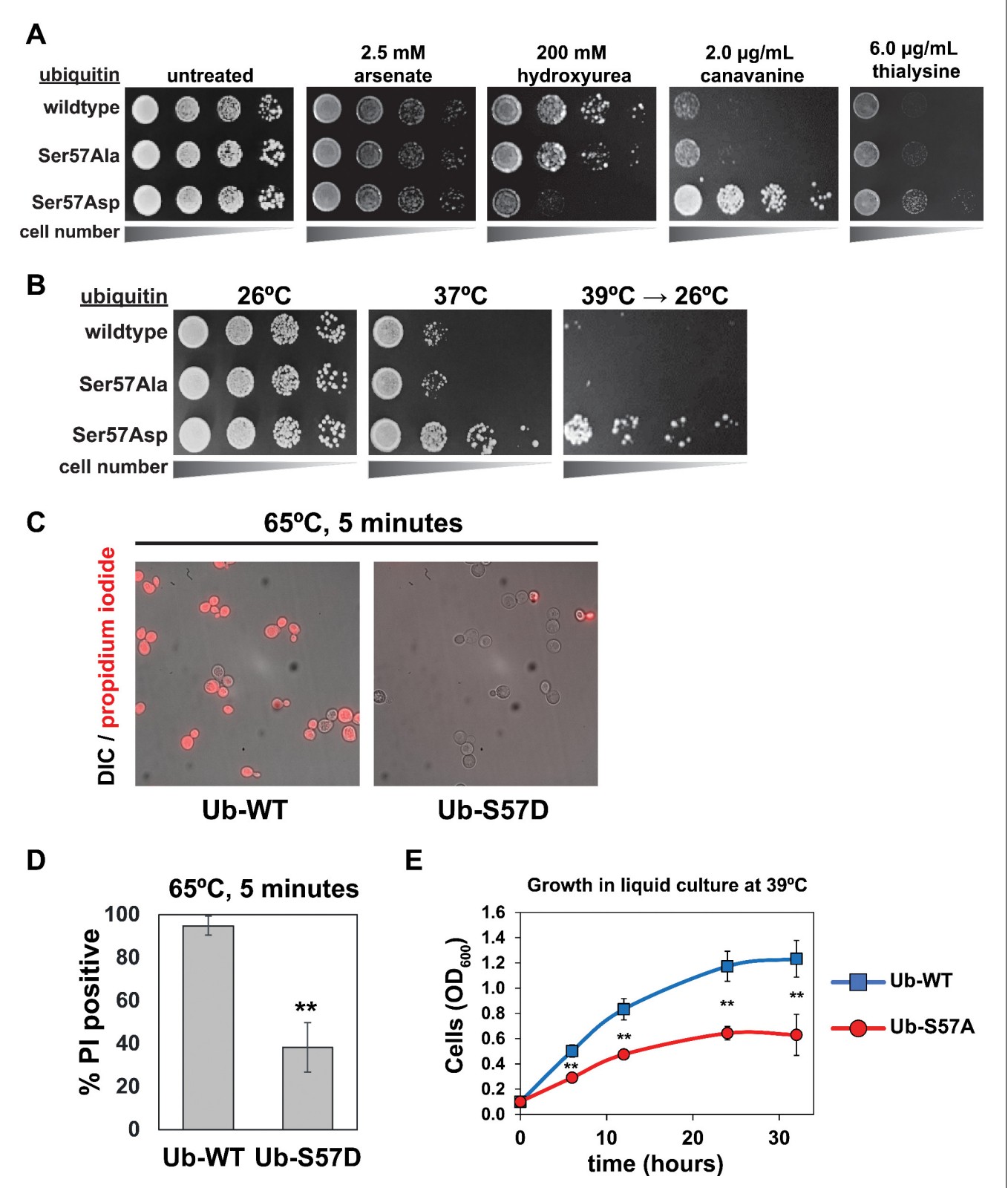

**Figure 1.** Genetic evidence of a role for Ser57 phosphorylation in the cellular response to stress. Ubiquitin variants were shuffled into the SUB280 yeast background (as described in the Materials and Methods) to generate strains that express exclusively wildtype, S57A, or S57D ubiquitin. (**A**) Yeast strains expressing the indicated ubiquitin variants (wildtype, S57A or S57D) were plated in serial dilutions onto the indicated synthetic dextrose medium (SDM) agar plates. (**B**) Serial dilutions of yeast cells were plated onto YPD agar plates and incubated at 26°C, 37°C, or 39°C for 18 hr and shifted back to
*Figure 1 continued on next page*

Figure 1 continued

26°C for recovery. (C) Fluorescence microscopy and (D) quantification of cells stained with propidium iodide (PI) after incubation at 65°C for 5 min. Dead cells are stained with PI. Results are means (n = 3) of % PI-positive cells ± SD (error bars). Double asterisk (**) indicates p value < 0.005 using a Student's t-test. (E) With a starting $OD_{600}$ of 0.1, cells were grown in liquid culture (-URA synthetic medium) at 39°C and $OD_{600}$ was measured at different time points. Data was collected from four biological replicate experiments (n=4) and the mean $OD_{600}$ ± SD (error bars) was calculated for each time point. Double asterisk (**) indicates p value < 0.005 using a Student's t-test. For all statistical analysis associated with *Figure 1*, p values are reported in the *Figure 1—source data 1*.

The online version of this article includes the following source data and figure supplement(s) for figure 1:

**Source data 1.** Quantification and statistical analysis for viability stain measurements (*Figure 1D*) and cell growth measurements (*Figure 1E* and *Figure 1—figure supplement 1*).

**Figure supplement 1.** Yeast cells expressing S57A ubiquitin are sensitive to thermal stress.

response to oxidative stress (*Figure 2C–D*). Since substitutions at the Ser57 position of ubiquitin (S57A, S57D) might interfere with the recognition of ubiquitin polymers by linkage-specific antibodies, we used SILAC-MS to analyze how the expression of phosphomimetic (S57D) ubiquitin affects ubiquitin polymer linkage abundance compared to wildtype ubiquitin during conditions of oxidative stress. Consistent with the immunoblot results, this analysis revealed that S57D expression increases the production of K48-linked polymers by 39% during oxidative stress (*Figure 2—figure supplement 1* and *Supplementary file 1*). (Notably, K63-linked polymers are a blind spot of this analysis, since the K63-linked ubiquitin remnant peptide also harbors the Ser57 position which is mutated to Asp in the light channel of this experiment.) It is worth noting that complete ubiquitin replacement in these cells exaggerates the impact physiological phosphorylation of ubiquitin is likely to have on global poly-ubiquitin linkage abundance. Indeed, we found that oxidative stress induced an approximately two-fold increase in phosphorylation of ubiquitin at the Ser57 position (*Figure 2F* and *Figure 2—figure supplement 2*). Based on previous measurements (*Swaney et al., 2015*; *Lee et al., 2017*), this stress-induced ubiquitin phosphorylation remains sub-stoichiometric and may have localized effects but is unlikely to alter global poly-ubiquitin linkage patterns. Ultimately, a deeper understanding of how Ser57 phosphorylated ubiquitin contributes to cellular stress responses will require the identification and characterization of the kinases that produce it.

To identify candidate ubiquitin kinases, we screened for Ser57 phosphorylation activity by co-expressing ubiquitin and yeast kinases in *E. coli* and immunoblotting lysates using an antibody specific for Ser57 phosphorylated ubiquitin. Initially, we focused on candidate kinases for which mutants exhibit phenotypes corresponding to those observed for cells expressing S57A or S57D ubiquitin. We found that co-expression of ubiquitin with the kinase Vhs1 resulted in immunodetection of Ser57 phosphorylated ubiquitin (*Figure 3A* and *Figure 3—figure supplement 1*). Vhs1 is part of the yeast family of Snf1-related kinases (*Tumolo et al., 2020*), and additional screening revealed three other kinases in this family that phosphorylated ubiquitin at the Ser57 position: Sks1 (which is 43% identical to Vhs1) (*Figure 3B*), Gin4 and Kcc4 (*Figure 3—figure supplement 2*). A previous study reported consensus phosphorylation motifs for Vhs1, Gin4, and Kcc4, and all bear similarity to the amino acid sequence surrounding Ser57 in ubiquitin (*Supplementary file 2*; *Mok et al., 2010*).

We further characterized the activity of Vhs1 and Sks1. Analysis of Vhs1 and Sks1 activity using Phos-tag acrylamide gels (*Figure 3—figure supplement 3*) confirmed the production of Ser57 phosphorylated ubiquitin. Using purified recombinant Vhs1 and Sks1, we reconstituted kinase activity toward Ser57 of ubiquitin and found that both kinases exhibited a preference for polymers (tri-ubiquitin > di ubiquitin > mono ubiquitin) although the activity of Sks1 was noticeably greater than that of Vhs1 (*Figure 3C–D* and *Figure 3—figure supplement 4–5*). Analysis of linkage specificity in the phosphorylation reaction revealed that Vhs1 is active toward linear (M1-linked), K29-, and K48-linked tetramers, while Sks1 is active toward linear, K48-, and K63-linked polymers (*Figure 3—figure supplement 6*). These differences in linkage-specific activities in vitro may underlie non-overlapping functions for these two kinases in vivo. To test if the observed in vitro activity correlated with in vivo activity, we used SILAC to quantify changes to ubiquitin modifications following over-expression of Vhs1 or Sks1. Importantly, we observed that overexpression of either Vhs1 or Sks1 increased ubiquitin phosphorylation at the Ser57 position (*Figure 3E–F* and *Figure 3—figure supplement 7–8*). Interestingly, this analysis also revealed a number of phosphopeptides derived from Vhs1 within its kinase domain (Ser86) and its Ser-rich C-terminus (*Figure 3—figure supplement 9*), suggesting that

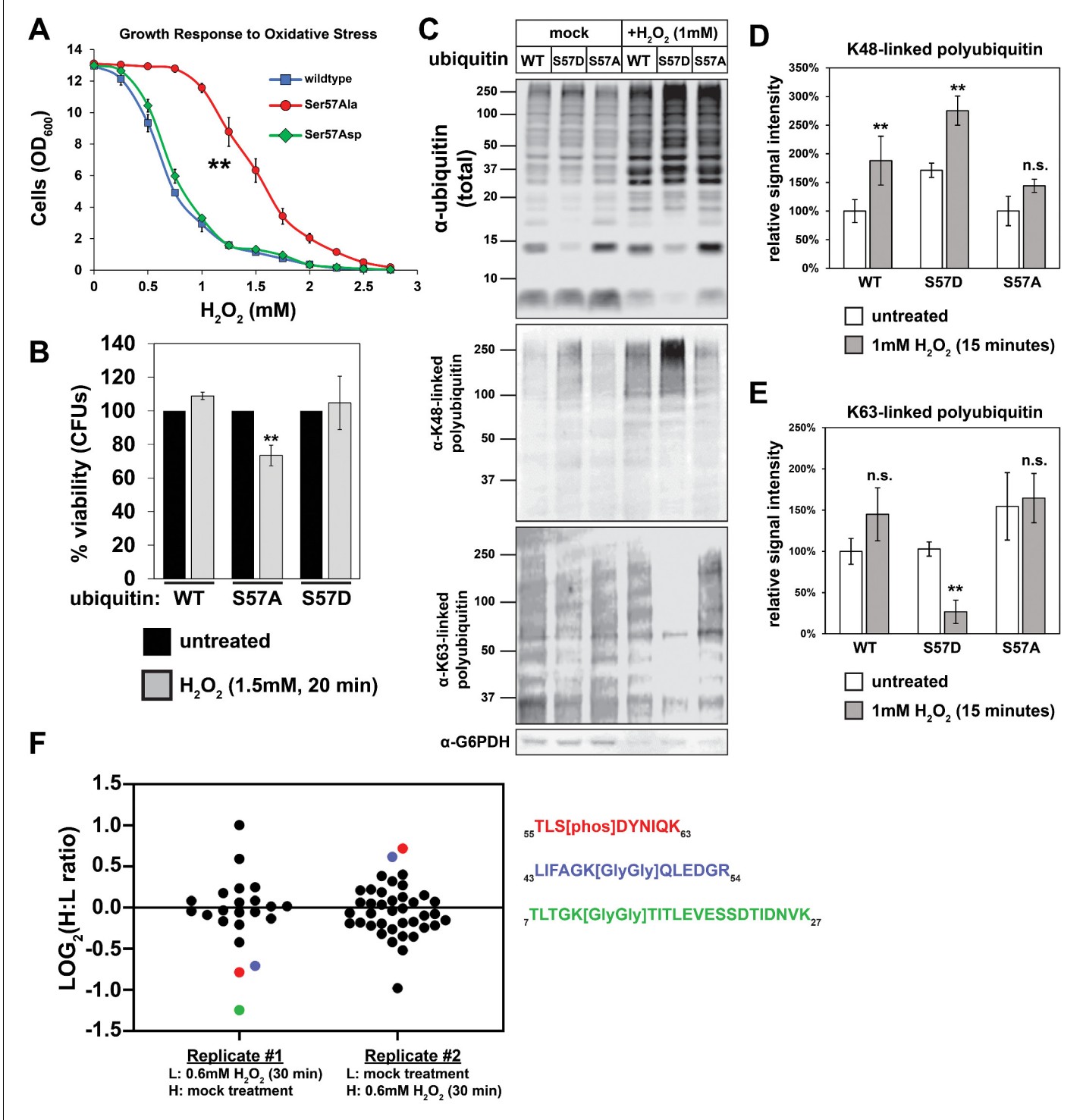

**Figure 2.** Ser57 phosphorylation of ubiquitin is important for the yeast oxidative stress response. (A) $OD_{600}$ of yeast cells cultured in YPD with different concentrations of $H_2O_2$ for 48 hr (starting $OD_{600}$ = 0.025). Data was collected for four biological replicate experiments (n=4) and the means ± SD (error bars) were calculated. Double asterisk (**) indicates p value < 0.005 between the range of 0.5 mM to 2.5 mM $H_2O_2$ using a Student's t-test, which was only observed for yeast expressing S57A ubiquitin. (B) CFU count of cells treated with 1.5 mM $H_2O_2$ for 20 min. Untreated control was taken before $H_2O_2$ treatment. Data was collected for three bioloigcal replicate experiments (n = 3) and the means ± SD (error bars) were calculated. Double asterisk (**) indicates p value < 0.05 comparing untreated to $H_2O_2$ treated cells using a Student's t-test, which revealed that only yeast expressing S57A ubiquitin experienced a significant loss of viability during $H_2O_2$ treatment in this experiment. (C) Western blot with anti-ubiquitin, anti-K48 ubiquitin, and anti-K63 ubiquitin antibodies of lysates from cells treated with 1 mM $H_2O_2$ for 30 min. (D and E) Quantification of results from the experiment

*Figure 2 continued on next page*

*Figure 2 continued*

shown in (C) is presented as the mean of four biological replicate experiments (n = 4) ± SD (error bars). Double asterisk (**) indicates p value < 0.005 comparing untreated to $H_2O_2$ treated cells using a Student's t-test. Correspondingly, p values > 0.05 are labeled as not significant (n.s.). Cells used in A-E are SUB280-derived strains exclusively expressing WT, S57A, or S57D ubiquitin variants. (F) SILAC-MS of affinity purified 3XFLAG-ubiquitin from yeast cells (JMY1312) treated with 0.6 mM $H_2O_2$ for 30 minutes. The peptide corresponding to Ser57 phosphorylation is colored red, while peptides corresponding to K48- and K11-linked poly-ubiquitin are colored blue and green, respectively. All other ubiquitin-derived peptide measurements are represented with black dots. All individual measurements and p values derived from statistical tests associated with the data in this figure are reported in *Figure 2—source data 1*.

The online version of this article includes the following source data and figure supplement(s) for figure 2:

**Source data 1.** Quantification and statistical analysis for the growth (*Figure 2A*), cell viability (*Figure 2B*), and anti-K48/K63 western blot analysis (*Figure 2D–E*) of cells in the presence or absence of $H_2O_2$.

**Figure supplement 1.** SILAC-MS fragmentation spectra of the K48-linked ubiquitin peptide isolated from yeast cells grown in the presence (light) or absence (heavy) of $H_2O_2$.

**Figure supplement 2.** SILAC-MS fragmentation spectra of unmodified (top) and Ser57-phosphorylated (bottom) peptides of ubiquitin isolated from yeast cells grown in the presence (light) or absence (heavy) of $H_2O_2$.

Vhs1 itself may be subject to complex phospho-regulation. Taken together our data indicate that Vhs1 and Sks1 are bona fide ubiquitin kinases that specifically phosphorylate the Ser57 position.

The family of Snf1-related kinases in yeast share homology with the human AMPK-related kinases (*Tumolo et al., 2020*). To test if human AMPK-related kinases exhibit activity toward ubiquitin, we performed in vitro kinase assays and found that a subset of this family phosphorylated ubiquitin at the Ser57 position specifically on tetramers (*Figure 3G*). This activity was apparent in the MARK kinases (MARK1-4) as well as related kinases SIK1 and SIK2. It is noteworthy that other candidate human ubiquitin kinases (initially identified by commercial screening services) did not exhibit Ser57 ubiquitin kinase activity in vitro (*Figure 3—figure supplement 10*). Mass spectrometry analysis confirmed production of Ser57 phosphorylated ubiquitin by MARK2 in vitro (*Figure 3—figure supplement 11*). Further analysis of MARK2 activity toward linkage-specific ubiquitin tetramers revealed a preference for linear, K11-, K29-, and K63-linked tetramers (*Figure 3—figure supplement 12*). Given that Ser57 ubiquitin phosphorylation activity was detected within yeast and human Snf1-related kinases (*Figure 3—figure supplement 13*), we propose that this is an evolutionarily conserved function for a subset of kinases within this family.

In an effort to understand the biological functions of Ser57-phosphorylated ubiquitin in yeast, we examined whether the deletion or overexpression of Ser57 ubiquitin kinases phenocopies expression of S57A or S57D ubiquitin, respectively (*Figure 1A–B* and *Figure 2A*). We did not observe heat tolerance phenotypes associated with deletion or overexpression of Ser57 ubiquitin kinases (*Figure 4—figure supplement 1–2*). However, yeast cells overexpressing *VHS1* exhibited resistance to canavanine and thialysine (*Figure 4A–B* and *Figure 4—figure supplement 3*), reminiscent of canavanine and thialysine resistance conferred by S57D expression (*Figure 1A*; *Lee et al., 2017*). Overexpression of a catalytic dead variant (*vhs1-K41R*, which harbors a mutation in the conserved ATP binding pocket of the kinase domain) did not confer canavanine or thialysine resistance (*Figure 4A–B* and *Figure 4—figure supplement 3*). Importantly, the canavanine and thialysine resistance associated with *VHS1* overexpression was suppressed in the presence of S57A ubiquitin (*Figure 4C–D* and *Figure 4—figure supplement 4*), indicating the phenotypes are driven by the production of Ser57-phosphorylated ubiquitin. We also observed that yeast cells overexpressing *SKS1* exhibited hypersensitivity to hydroxyurea (*Figure 4—figure supplement 5*), consistent with the hydroxyurea sensitivity phenotype observed in yeast cells expressing S57D ubiquitin (*Figure 1A*). The hydroxyurea hypersensitivity associated with *SKS1* overexpression was suppressed in the presence of S57A ubiquitin (*Figure 4E*), indicating the phenotype requires Ser57 phosphorylation of ubiquitin. Taken together, these phenotypic correlations suggest that Vhs1 and Sks1 exert stress phenotypes associated with the phosphorylation of ubiquitin at the Ser57 position.

We also examined whether the deletion of Ser57 ubiquitin kinases phenocopied the oxidative stress response defect observed for yeast cells expressing S57A ubiquitin (*Figure 2A*). Notably, Δsks1Δvhs1 double mutant cells failed to arrest growth in response to oxidative stress, a phenotype that could only be complemented by the re-introduction of both VHS1 and SKS1 (*Figure 4F*). This phenotype of Δsks1Δvhs1 double mutant cells was also suppressed by the expression of S57D (but

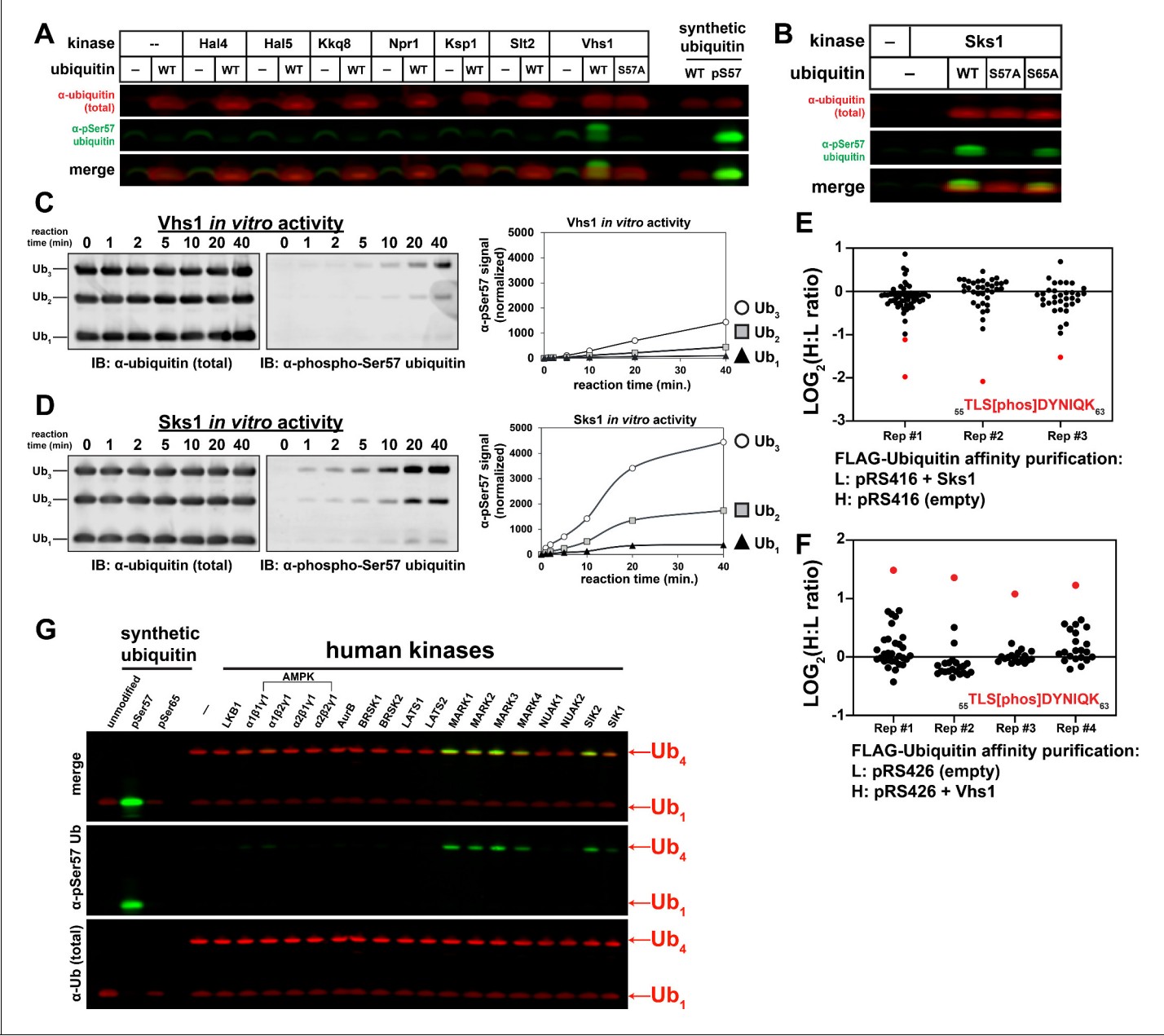

**Figure 3.** A subset of the Snf1-related family of kinases phosphorylates ubiquitin at the Ser57 position. (**A**) Anti-phospho-Ser57 western blot of *E. coli* Rosetta 2 (DE3) lysates co-expressing ubiquitin and yeast kinases. (**B**) Anti-phospho-Ser57 western blot of *E. coli* Rosetta 2 (DE3) lysates co-expressing ubiquitin (wildtype, S57A, or S65A variants) and Sks1, a paralog of Vhs1. (**C and D**) In vitro reconstitution of ubiquitin Ser57 phosphorylation using purified recombinant Vhs1 (**C**) and Sks1 (**D**). Ubiquitin monomers as well as linear (M1-linked) dimers and trimers were included in equal amounts. Numerical values above each lane represent reaction time for the sample. (**E and F**) SILAC-MS of IP-enriched 3XFLAG ubiquitin from yeast cells (JMY1312 background) with either empty vector or with vector for overexpression of Sks1 (**E**) or Vhs1 (**F**). Black and red dots indicate resolved ubiquitin peptides and Ser57-phosphopeptides, respectively. All individual measurements are reported in *Figure 3—source data 1*. (**G**) In vitro activity assay of select human kinases of the AMPK family using equal amounts of mono-ubiquitin and M1-linked tetra-ubiquitin as substrates.

The online version of this article includes the following source data and figure supplement(s) for figure 3:

**Source data 1.** Source measurements for the H:L ratio of ubiquitin peptides resolved in all replicate experiments of SUB280 cells overexpressing Sks1 (*Figure 3E*) or Vhs1 (*Figure 3F*).

**Figure supplement 1.** Anti-phospho-Ser57 western blot of *E.coli* Rosetta 2 (DE3) lysates co-expressing ubiquitin (wildtype, S57A, or S65A) and Vhs1.

**Figure supplement 2.** Anti-phospho-Ser57 western blot of *E.coli* Rosetta 2 (DE3) whole-cell lysates after heterologous co-expression of ubiquitin and yeast members of the Snf1-related family of kinases.

**Figure supplement 3.** Phos-tag gel separation of ubiquitin (wildtype, S57A, or S65A variants) following co-expression with Vhs1 or Sks1.

*Figure 3 continued on next page*

*Figure 3 continued*

**Figure supplement 4.** Detection of Ser57 phosphorylation of ubiquitin in the presence of Vhs1 by mass spectrometry.
**Figure supplement 5.** Detection of Ser57 phosphorylation of ubiquitin in the presence of Sks1 by mass spectrometry.
**Figure supplement 6.** Vhs1 and Sks1 have distinct linkage-type preferences.
**Figure supplement 7.** Overexpression of Sks1 in yeast cells increases Ser57 phosphorylation of ubiquitin.
**Figure supplement 8.** Overexpression of Vhs1 in yeast cells increases Ser57 phosphorylation of ubiquitin.
**Figure supplement 9.** Phosphopeptides detected from Vhs1.
**Figure supplement 10.** Screening of human ubiquitin Ser57 kinases by Western Blot.
**Figure supplement 11.** Identification and characterization of human MARK2 phosphorylation site on ubiquitin.
**Figure supplement 12.** Determination of MARK2 substrate preference by analyzing activity toward tetra-ubiquitin chains with different linkages.
**Figure supplement 13.** Yeast ubiquitin kinases Sks1, Vhs1, Gin4, and Kcc4 cluster with human MARKs of the CAMK superfamily of kinases.

not WT or S57A) ubiquitin (*Figure 4G*) suggesting that the growth arrest defect in Δ*sks1*Δ*vhs1* cells is linked to a deficiency in the production of ubiquitin phosphorylated at the Ser57 position. To explore this further, we used SILAC-MS to compare levels of Ser57 phosphorylated ubiquitin in untreated or $H_2O_2$-treated cells, however, this analysis did not reveal significant changes in Δ*sks1*Δ*vhs1* cells (*Figure 4H* and *Table 1*). While these results indicate that Sks1 and Vhs1 are dispensable for production of Ser57 phosphorylated ubiquitin in the acute phase of the oxidative stress response, we cannot exclude roles for these kinases during prolonged exposure to oxidative stress or the possibility that they contribute to the phosphorylation of localized pools of ubiquitin.

## Discussion

The data presented here provide evidence that Ser57 phosphorylated ubiquitin and the kinases that produce it play an important role in several yeast stress responses, but the low stoichiometry of this modification suggests its effects are likely limited and localized in a physiological context. To the best of our knowledge, this study reports the first Ser57 ubiquitin kinases, and the only known ubiquitin kinases besides PINK1. Importantly, PINK1 activity is tightly regulated and highly localized to the outer membrane of damaged mitochondria. One potential explanation of the genetic and biochemical data presented here is that Sks1 and Vhs1 phosphorylate a localized pool (or pools) of ubiquitin in response to stress. However, the data also indicate that Sks1 and Vhs1 are not required for the production of Ser57 phosphorylated ubiquitin during normal growth or oxidative stress (*Figure 4H* and *Table 1*). This may be due to redundancy of kinase activities, possibly with Gin4, Kcc4, or other as-yet-unidentified ubiquitin kinases. However, such redundancy cannot explain the phenotypes observed during oxidative stress, since Δ*sks1*Δ*vhs1* double mutant cells exhibit oxidative stress phenotypes that are suppressed by expression of phosphomimetic (S57D) ubiquitin (*Figure 4G*). In this case, we propose that a localized pool of ubiquitin phosphorylated by Sks1 and Vhs1 is critical for the oxidative stress response but is not resolved in our proteomic analysis, or only contributes a small fraction to a larger pool. This interpretation reconciles the genetic and biochemical data presented here, and it is consistent with the precedent established with PINK1, which is subject to tight regulation and contributes to localized production of Ser65 phosphorylated ubiquitin. However, further characterization of the Ser57 ubiquitin kinases reported here and analysis of localized ubiquitin pools will be required to validate this interpretation.

Genetic experiments presented here reveal that ubiquitin replacement with phosphomimetic ubiquitin phenocopies kinase gain of function (as is the case with canavanine, thialysine, and hydroxyurea sensitivities) while replacement with phosphorylation resistant ubiquitin phenocopies kinase loss of function (as is the case with sensitivity to $H_2O_2$). An important limitation of this genetic analysis is the built-in assumption that ubiquitin variants (phosphomimetic or phosphorylation resistant) behave as expected and do not alter other biochemical properties of ubiquitin in a cellular context. Additionally, such phenotypes associated with ubiquitin variants are likely to over-estimate the physiological effects of ubiquitin phosphorylation. In the case of phospho-mimetic (S57D) ubiquitin, this is because physiological ubiquitin phosphorylation is sub-stoichiometric. In the case of phosphorylation resistant (S57A) ubiquitin, this may be due to redundancy of kinases. Such redundancy seems likely since we identified multiple Ser57 ubiquitin kinases (Vhs1, Sks1, Gin4, and Kcc4) and the deletion of two kinases did not decrease the production of Ser57 phosphorylated ubiquitin

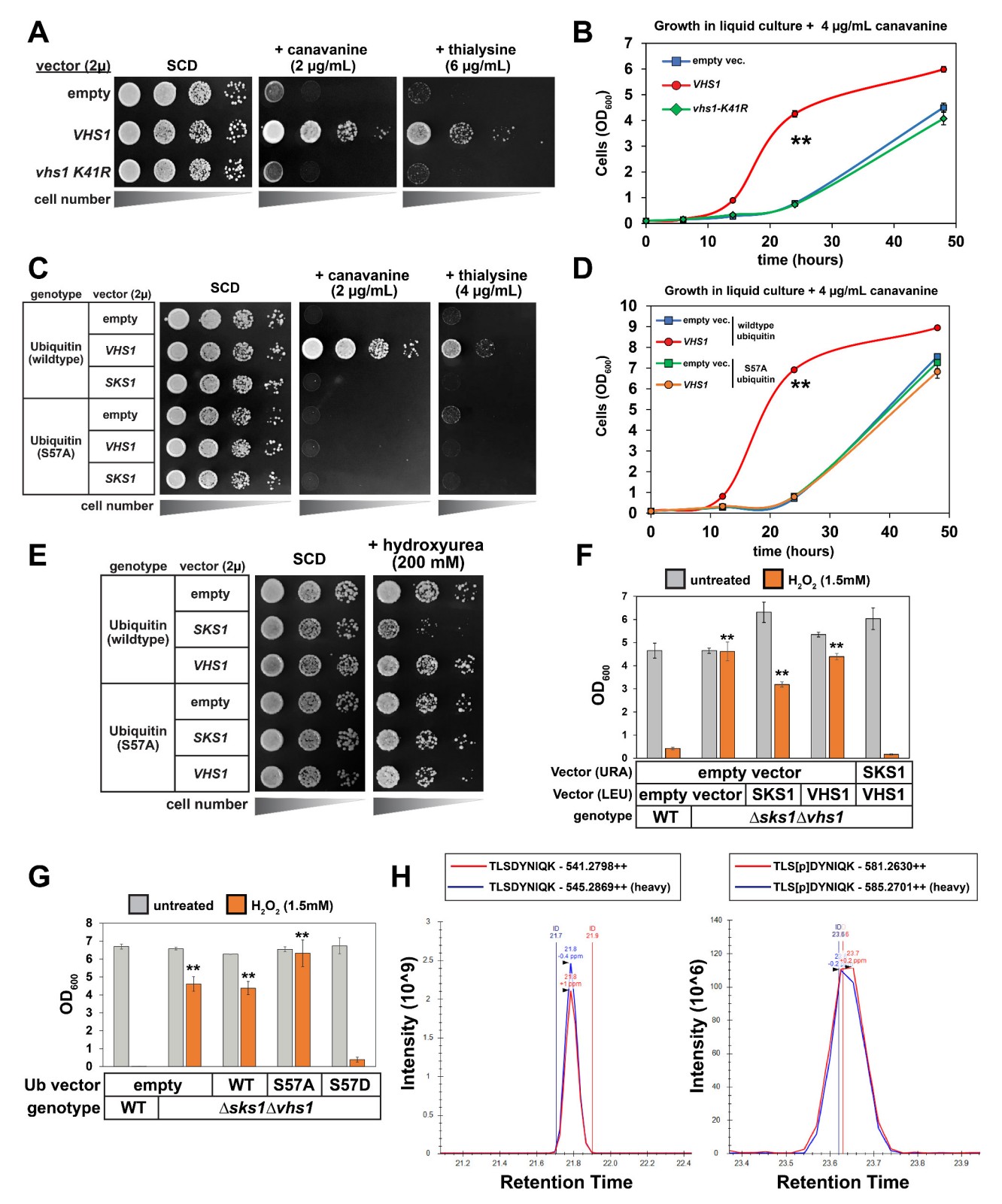

**Figure 4.** Regulation of proteotoxic stress responses by Ser57 ubiquitin kinases. (**A**) Analysis of yeast strains containing high copy plasmids (empty vector, *VHS1*, or a kinase dead (*vhs1 K41R*) variant) grown on the indicated synthetic media plates. (**B**) Analysis of the same yeast strains in (**A**) grown in liquid culture containing canavanine (4 μg/mL). Values in each time point are means of three biological replicates (n = 3) and ± SD (error bars). Double asterisk (**) indicates p value < 0.005 at the 14 and 24 hr time points using a Student's t-test, which was only observed for yeast overexpressing

*Figure 4 continued on next page*

*Figure 4 continued*

wildtype *VHS1*. (C) Analysis of yeast strains (SUB280 background, expressing either WT or S57A ubiquitin) containing the indicated high copy plasmids (empty vector or harboring *VHS1* or *SKS1*) grown on the indicated synthetic media plates. (D) Analysis of the same yeast strains in (C) grown in liquid culture containing canavanine (4 μg/mL). Values in each time point are means of three biological replicates (n = 3) and ± SD (error bars). Double asterisk (\*\*) indicates p value < 0.005 at the 14 and 24 hr time points using a Student's t-test, which was only observed for yeast overexpressing wildtype *VHS1* and expressing wildtype ubiquitin. (E) Analysis of yeast strains (SUB280 background, expressing either WT or S57A ubiquitin) containing the indicated high copy plasmids (empty vector or harboring *VHS1* or *SKS1*) grown on the indicated synthetic media plates. (F) Analysis of the growth response to oxidative stress in WT and Δ*sks1*Δ*vhs1* double mutant cells, with indicated complementation expression vectors. $OD_{600}$ of cells (n = 4; ± SD error bars) in dropout SM media in the absence or presence of 1.5 mM $H_2O_2$ after incubation at 24 hr or 48 hr, respectively. Starting $OD_{600}$ of 0.025 and data was collected for four biological replicate experiments (n=4). Double asterisk (\*\*) indicates p value < 0.001 compared to wildtype yeast treated with $H_2O_2$ using a Student's t-test, which was observed for yeast expressing empty vector, *VHS1* individually, or *SKS1* individually but not for yeast expressing both *VHS1* and *SKS1*. (G) Analysis of the growth response to oxidative stress in WT and Δ*sks1*Δ*vhs1* double mutant cells, with indicated ubiquitin expression vectors. $OD_{600}$ of cells (n = 4; ± SD error bars) in dropout SM media in the absence or presence of 1.5 mM $H_2O_2$ after incubation at 24 hr or 48 hr, respectively. Starting $OD_{600}$ of 0.025 and data was collected for four biological replicate experiments (n=4). Double asterisk (\*\*) indicates p value < 0.001 compared to wildtype yeast treated with $H_2O_2$ using a Student's t-test, which was observed for yeast expressing empty vector, wildtype ubiquitin, or S57A ubiquitin but not S57D ubiquitin. (H) Chromatography data serving as the basis for quantification of Ser57 phosphorylation corresponding to *Table 1*. Yeast cells expressing endogenous FLAG-ubiquitin (JMY1312 [SEY6210 background] parent cells) were cultured in heavy (H; wildtype cells [blue trace]) or light (L, isogenic Δ*vhs1*Δ*sks1* deletion [red trace]) SILAC media to the mid-log phase and cells were treated with 1 mM $H_2O_2$ for 30 min before harvesting. Following cell lysis and overnight tryptic digestion of affinity-purified FLAG-ubiquitin, phospho-peptides were enriched using IMAC chromatography (see Materials and Methods) and analyzed by mass spectrometry. The Ser57 phosphopeptide is represented in the right panel, while the corresponding unmodified peptide is depicted in the left panel. All individual measurements and p values derived from statistical tests associated with the data in this figure are reported in *Figure 4—source data 1*.

The online version of this article includes the following source data and figure supplement(s) for figure 4:

**Source data 1.** Quantification and statistical analysis for growth tolerance in canavanine (*Figure 4B and D*), thialysine (*Figure 4—figure supplements 3* and *4*), and hydrogen peroxide (*Figure 4E and F*).

**Figure supplement 1.** Heat stress growth phenotype of yeast strains overexpressing Ser57 ubiquitin kinases.

**Figure supplement 2.** Analysis of cell tolerance to heat stress in yeast kinase deletion cells.

**Figure supplement 3.** Analysis of cell tolerance to thialysine of yeast cells overexpressing *VHS1* or a kinase dead (*vhs-K41R*) variant (kinase dead).

**Figure supplement 4.** Analysis of cell tolerance to thialysine of yeast cells overexpressing *VHS1*.

**Figure supplement 5.** Hydroxyurea growth phenotype of yeast strains overexpressing Ser57 ubiquitin kinases.

(*Figure 4H* and *Table 1*). Furthermore, we cannot exclude the possibility that mutation of Ser57 alters the biochemical properties of ubiquitin in such a way as to phenocopy the effects associated with kinase deletion and/or overexpression. Going forward, these limitations present formidable challenges for dissecting the biological functions of ubiquitin phosphorylation, which is why the identification and characterization of ubiquitin kinases is critical. Ultimately, deeper

**Table 1.** Corresponds to *Figure 4H*.

Analysis of ubiquitin phosphorylation in Δ*vhs1*Δ*sks1* mutants. Yeast cells expressing endogenous FLAG-ubiquitin (JMY1312 [SEY6210 background] parent cells) were cultured in heavy (H; wildtype cells) or light (L, isogenic Δ*vhs1*Δ*sks1* deletion) SILAC media to the mid-log phase. In Experiment #1, cells were untreated, and in Experiment #2 cells were treated with 1 mM $H_2O_2$ for 30 min before harvesting cells. Following cell lysis and overnight digestion of lysates with trypsin, phospho-peptides were enriched using IMAC chromatography (see Materials and methods) and analyzed by mass spectrometry. Phosphorylated ubiquitin peptides resolved in these experiments are indicated and the values presented in the table represent the H:L SILAC ratio, which has been normalized to the total ubiquitin H:L ratio for each experiment and log-transformed ($log_2$). 'n.d.' for Ser19 in Experiment #1 indicates that this peptide was identified but could not be quantified in this experiment due to poor quality of chromatographic data.

| Peptide | Phospho- site | $log_2$ (H:L Ratio) | |
| --- | --- | --- | --- |
| | | Expt #1 | Expt #2 |
| TITLEVE[pS]SDTIDNVK | Ser19 | n.d. | 0.18 |
| TL[pS]DYNIQK | Ser57 | 0.06 | −0.06 |
| ES[pT]LHLVLR | Thr66 | −0.22 | 0.11 |

characterization of Ser57 ubiquitin kinases – particularly to understand their localization and regulation in the context of proteotoxic stress – will likely be critical to understanding how phosphorylation regulates the biology of the ubiquitin code.

# Materials and methods

## Key resources table

| Reagent type (species) or resource | Designation | Source or reference | Identifiers | Additional information |
|---|---|---|---|---|
| Strain, strain background (*S. cerevisiae*) | SUB280 | D. Finley Lab; *Finley et al., 1994* PMID:8035826 PMCID:PMC359070 | | MATa, *lys2-801, leu2-3, 112, ura3 -52, his3-D200, trp 1–1, ubi1-D1::TRP1, ubi2-D2:: ura3, ubi3-Dub-2, ubi4-D2::LEU2* [*pUB39 Ub, LYS2*] [*pUB100, HIS3*] |
| Strain, strain background (*S. cerevisiae*) | SUB280 Ub (WT, S57A, S57D) | *Lee et al., 2017* PMID:29130884 PMCID:PMC5706963 | | MATa, SUB280 [*RPS31* (WT, S57A, S57D; *URA3*)] |
| Strain, strain background (*S. cerevisiae*) | NHY318 | This paper | | MATa, SUB280 *arg4::KANMX* |
| Strain, strain background (*S. cerevisiae*) | SEY6210 | S. Emr Lab; *Robinson et al., 1988* PMID:3062374 PMCID:PMC365587 | RRID:ATCC: 96099 | WT: MATα leu2-3,112 ura4-52 his3-D200 trp1-D901 lys2-801 suc2-D9 |
| Strain, strain background (*S. cerevisiae*) | SEY6210.1 | S. Emr Lab; *Robinson et al., 1988* PMID:3062374 PMCID:PMC365587 | | WT: MATa leu2-3,112 ura4-52 his3-D200 trp1-D901 lys2-801 suc2-D9 |
| Strain, strain background (*S. cerevisiae*) | NHY175 | This paper | | MATα, SEY6210 *gin4::NATMX* |
| Strain, strain background (*S. cerevisiae*) | NHY117 | This paper | | MATα, SEY6210 *sks1::TRP1* |
| Strain, strain background (*S. cerevisiae*) | NHY118 | This paper | | MATα, SEY6210 *vhs1::TRP1* |
| Strain, strain background (*S. cerevisiae*) | NHY196 | This paper | | MATa, SEY6210.1 *sks1::TRP1 vhs1::TRP1* |
| Strain, strain background (*S. cerevisiae*) | NHY177 | This paper | | MATα, SEY6210 *gin4::NATMX sks1::TRP1* |
| Strain, strain background (*S. cerevisiae*) | NHY179 | This paper | | MATα, SEY6210 *gin4::NATMX vhs1::TRP1* |
| Strain, strain background (*S. cerevisiae*) | NHY181 | This paper | | MATα, SEY6210 *gin4::NATMX sks1::TRP1 vhs1::TRP1* |
| Strain, strain background (*S. cerevisiae*) | JMY1312 | This paper | | MATα, SEY6210 *arg4::KANMX FLAG-RPS3::TRP1, FLAG-RPL40B::TRP1* |
| Strain, strain background (*S. cerevisiae*) | NHY326 | This paper | | MATa, JMY1312 *sks1::TRP1 vhs1::TRP1* |
| Strain, strain background (*E. coli*) | Rosetta 2 (DE3) | Millipore (Burlington, MA) | Cat# 71400 | F⁻ *ompT hsdS*$_B$(r$_B^-$ m$_B^-$) *gal dcm* (DE3) pRARE2 (Cam$^R$) Competent Cells - Novagen |

*Continued on next page*

*Continued*

| Reagent type (species) or resource | Designation | Source or reference | Identifiers | Additional information |
|---|---|---|---|---|
| Strain, strain background (*E. coli*) | C41 (DE3) | Lucigen Corporation (Middleton, WI) | | F⁻ ompT hsdSB (rB - mB -) gal dcm (DE3) Competent Cells |
| Recombinant DNA | pRS415 | Stratagene | | (Empty Backbone) Yeast Centromere Plasmid (LEU2) GenBank: U03449 |
| Recombinant DNA | pRS416 | *Sikorski and Hieter, 1989* | | (Empty Backbone) Yeast Centromere Plasmid (URA3) GenBank: U03450 |
| Recombinant DNA | pJAM1303 | This paper | | pNative-Sks1 in pRS416 (URA3); yeast expression |
| Recombinant DNA | pJAM1304 | This paper | | pNative-Sks1 in pRS415 (LEU2); yeast expression |
| Recombinant DNA | pJAM1280 | This paper | | pTDH3-Sks1 in pRS415 (LEU2); yeast expression |
| Recombinant DNA | pJAM1253 | This paper | | pNative_Vhs1 in pRS415 (LEU2); yeast expression |
| Recombinant DNA | pRS327 | PMID:14989082 | RRID:Addgene_51787 | (Empty Backbone) multicopy (YEp) vector with a 2 µm origin of replication (LYS2); yeast expression |
| Recombinant DNA | pJAM1746 | This paper | | pTDH3-Gin4 in pRS327; yeast expression |
| Recombinant DNA | pJAM1749 | This paper | | pTDH3-Kcc4 in pRS327; yeast expression |
| Recombinant DNA | pJAM1743 | This paper | | pTDH3-Sks1 in pRS327; yeast expression |
| Recombinant DNA | pJAM1740 | This paper | | pTDH3-Vhs1 in pRS327; yeast expression |
| Recombinant DNA | pJAM1771 | This paper | | pTDH3-Vhs1 K41R (kinase dead) in pRS327; yeast expression |
| Recombinant DNA | pJAM1240 | This paper | | 6XHis-Sks1 in pET15b, AmpR; *E. coli* expression |
| Recombinant DNA | pJAM1169 | This paper | | Vhs1-6XHis in pET23d, AmpR; *E. coli* expression |
| Recombinant DNA | pJAM983 | This paper | | Gin4-6XHis in pET23d, AmpR; *E. coli* expression |
| Recombinant DNA | pJAM1116 | This paper | | Hal4-6XHis in pET23d, AmpR; *E. coli* expression |
| Recombinant DNA | pJAM1118 | This paper | | Hal5-6XHis in pET23d, AmpR; *E. coli* expression |
| Recombinant DNA | pJAM1120 | This paper | | Kkq8-6XHis in pET23d, AmpR; *E. coli* expression |
| Recombinant DNA | pJAM1336 | This paper | | Snf1-6XHis in pET23d, AmpR; *E. coli* expression |
| Recombinant DNA | pJAM585 | This paper | | 6XHis-Npr1 in pCOLADuet, KmR; *E. coli* expression |
| Recombinant DNA | pJAM1270 | This paper | | Ksp1-6XHis in pET23d, AmpR; *E. coli* expression |
| Recombinant DNA | pJAM1186 | This paper | | GST-Slt2 in pGEX6p1; *E. coli* expression |
| Recombinant DNA | pJAM1335 | This paper | | Frk1-6XHis in pET23d, AmpR; *E. coli* expression |

*Continued on next page*

*Continued*

| Reagent type (species) or resource | Designation | Source or reference | Identifiers | Additional information |
|---|---|---|---|---|
| Recombinant DNA | pJAM987 | This paper | | Kcc4-6XHis in pET23d, AmpR; *E. coli* expression |
| Recombinant DNA | pJAM1306 | This paper | | Kin1-6XHis in pET23d, AmpR; *E. coli* expression |
| Recombinant DNA | pJAM1307 | This paper | | Kin2-6XHis in pET23d, AmpR; *E. coli* expression |
| Recombinant DNA | pJAM1272 | This paper | | Ypl150w-6XHis in pET23d, AmpR; *E. coli* expression |
| Recombinant DNA | pJAM1167 | This paper | | 6XHis-GST-UBD, AmpR; *E. coli* expression |
| Recombinant DNA | pJAM1235 | This paper | | Ubiquitin in pBG100; *E. coli* expression |
| Recombinant DNA | pJAM1236 | This paper | | Ubiquitin S57A in pBG100; *E. coli* expression |
| Recombinant DNA | pJAM1237 | This paper | | Ubiquitin S65A in pBG100; *E. coli* expression |
| Recombinant DNA | pJAM995 | This paper | | di-Ub in pET23d, AmpR; *E. coli* expression |
| Recombinant DNA | pJAM996 | This paper | | tri-Ub in pET23d, AmpR; *E. coli* expression |
| Antibody | α-Flag (FLAG(R) M2) mouse monoclonal | Sigma (St. Louis, MO) | RRID:AB_262044 | WB (1:1000) |
| Antibody | α-Ubiquitin (VU101) mouse monoclonal; clone VU-1 | LifeSensors (Malvern, PA) | Cat# VU101 RRID:AB_2716558 | WB (1:1000) |
| Antibody | α-Glucose-6-Phosphate Dehydrogenase (G6PDH), yeast rabbit polyclonal | Sigma (St. Louis, MO) | Cat# A9521 RRID:AB_258454 | WB (1:10000) |
| Antibody | α-pSer57 Ubiquitin rabbit polyclonal | This paper | | WB (1:1000) |
| Antibody | K48-linkage Specific Polyubiquitin rabbit monoclonal; clone D9D5 | Cell Signaling Technology (Danvers, MA) | Cat# 8081 RRID:AB_10859893 | WB (1:1000) |
| Antibody | K63-linkage Specific Polyubiquitin rabbit monoclonal; clone Apu3 | Millipore (Burlington, MA) | Cat# 05–1308 RRID:AB_1587580 | WB (1:1000) |
| Antibody | IRDye 680RD-Goat anti-mouse polyclonal | LI-COR Biosciences (Lincoln, NE) | Cat# 926–68070 RRID:AB_10956588 | WB (1:10000) |
| Antibody | IRDye 800CW-Goat anti-rabbit polyclonal | LI-COR Biosciences (Lincoln, NE) | Cat# 926–32211 RRID:AB_621843 | WB (1:10000) |
| Commercial assay, kit | PTMScan Ubiquitin Remnant Motif (α-K-ε-GG) | Cell Signaling Technology (Danvers, MA) | RRID:Cat# 5562 | bead-conjugated for immunoprecipitation of –GG-εK-conjugated peptides |
| Chemical compound, drug | EZ view Red Anti-FLAG M2 Affinity Gel | Sigma (St. Louis, MO) | Cat# F2426-1ML RRID:AB_2616449 | mouse-monoclonal; clone M2 bead-conjugated for immuno precipitation of FLAG-conjugated proteins |
| Chemical compound, drug | L-Lysine-$^{13}C_6$,$^{13}N_2$ hydrochloride | Sigma (St. Louis, MO) | Cat# 608041–1G | |
| Chemical compound, drug | L-Arginine-$^{13}C_6$,$^{13}N_4$ hydrochloride | Sigma (St. Louis, MO) | Cat# 608033–1G | |

*Continued on next page*

*Continued*

| Reagent type (species) or resource | Designation | Source or reference | Identifiers | Additional information |
|---|---|---|---|---|
| Chemical compound, drug | Sodium arsenate dibasic heptahydrate | Sigma (St. Louis, MO) | Cat# A6756 | Synonym: arsenate |
| Chemical compound, drug | Hydrogen peroxide | Fisher (Hampton, NH) | Cat# H325-500 | Synonym: $H_2O_2$ |
| Chemical compound, drug | Hydroxyurea | INDOFINE Chemical Company, Inc (Hillsborough, NJ) | Cat# BIO-216 | |
| Chemical compound, drug | S-(2-Aminoethyl)-L-cysteine hydrochloride | Sigma (St. Louis, MO) | Cat# A2636-1G | Synonym: L-4-Thialysine hydrochloride, thialysine |
| Chemical compound, drug | L-Canavanine | Sigma (St. Louis, MO) | Cat. # C1625 | Synonym: canavanine |
| Chemical compound, drug | DL-2-Aminoadipic acid | Sigma (St. Louis, MO) | Cat. # A0637 | |
| Chemical compound, drug | MG132 | Apexbio (Houston, TX) | Cat. # A2585 | |
| Chemical compound, drug | Protease inhibitor cocktail (cOmplete Tablets, Mini) | Roche (Basel, Switzerland) | Cat# 04693159001 | |
| Chemical compound, drug | PhosSTOP | Roche (Basel, Switzerland) | Cat# 04906837001 | |
| Chemical compound, drug | Phos-tag Acrylamide | NARD | Cat# AAL-107 | |
| Software, algorithm | MaxQuant Version 1.6.5.0 | Max Planck Institute of Biochemistry | RRID:SCR_014485 | |
| Software, algorithm | Skyline Version 20.1.0.155 | MacCoss Lab Software | RRID:SCR_014080 | |
| Software, algorithm | Adobe Illustrator CS5.1 Version 15.1.0 | Adobe (San Jose, CA) | RRID:SCR_010279 | |
| Software, algorithm | Image Studio Lite Version 5.2 | LI-COR Biosciences (Lincoln, NE) | RRID:SCR_013715 | |

## Cell strains and culture

All *Saccharomyces cerevisiae* strains and reagents used in this study are described in the Key Resources Table. SUB280 cells were used to shuffle different ubiquitin variants (wildtype, S57A, and S57D) by counterselection on URA-dropout SDM plates containing DL-2-aminoadipic acid as previously described (*Lee et al., 2019*; *Sloper-Mould et al., 2001*). Such SUB280-derived cells were used in growth sensitivity assays in different stress conditions. SEY6210 cells were used to generate gene-deletion strains by resistance marker-guided recombination and cross-mating methods. Yeast strains for mass spectrometry experiments were JMY1312 (with chromosomally tagged *RPS31* and *RPL40B*) and its derivatives. Cells were cultured at 26°C in yeast-peptone-dextrose (YPD) or synthetic dextrose minimal medium (SDM: 0.67% yeast nitrogen base, 2% dextrose, and required amino acids) at 200 rpm agitation. SILAC media were supplemented with light lysine ($^{12}C_6^{14}N_2$ L-Lys) and arginine ($^{12}C_6^{14}N_4$ L-Arg), or heavy isotopes of lysine ($^{13}C_6^{15}N_2$ L-Lys) and arginine ($^{13}C_6^{15}N_4$ L-Arg). For spot plate dilution assay, cells were grown for 18–24 hr, normalized to $OD_{600}$ of 1.0, and sequentially diluted at 1:10 dilution onto SDM agar plates containing amino acid dropout mixture in the absence or presence of 2.5 mM $NaH_2AsO_4$, 200 mM hydroxyurea, 2.0 µg/mL canavanine, and 6.0 µg/mL thialysine. For $H_2O_2$ sensitivity assay, yeast cells grown to the mid-log phase ($OD_{600}$ of 0.7) were diluted to $OD_{600}$ of 0.025 with 1.5 mM $H_2O_2$ in SDM and terminal $OD_{600}$ of cells was measured after 1–3 days of incubation. For growth sensitivity assay of liquid cultures, cells were grown for 18 hr at 26°C, normalized to $OD_{600}$ of 5.0, and subcultured into fresh media with a starting $OD_{600}$ of 0.1. $OD_{600}$ of cells was recorded at different time points until cells reach the stationary phase. For bacterial cultures, the yeast kinase and ubiquitin were heterologously co-expressed in *E. coli* Rosetta 2 (DE3) by 1 mM IPTG induction at 26°C for 16 hr.

## Protein preparation

### Protein precipitation

Yeast proteins were precipitated by adding ice-cold 10% trichloroacetate in TE buffer (2 mM EDTA, 10 mM Tris-HCl, pH 8.0), washed with 100% acetone, and lyophilized by vacuum centrifugation. Desiccated protein was resolubilized in urea sample buffer.

### Recombinant protein expression and purification of kinases

N-terminally tagged Vhs1 or Sks1 were produced in *E. coli* C41(DE3) cells cultured in LB Media. Cells were induced at an $OD_{600}$ of 0.6 with 1 mM IPTG and allowed to express for 4 hr at 37°C. Cells were pelleted by centrifugation at 6000 $\times$ *g* for 25 min and flash-frozen with liquid nitrogen. Before purification, the cells were thawed on ice, resuspended in 5 mL of lysis buffer (50 mM Tris pH 8.0, 150 mM NaCl, 10 mM imidazole, 2 mM βME, complete protease inhibitors [Roche, Basel, Switzerland], 1 μg/mL DNase, 1 μg/mL lysozyme, and 1 mM PMSF) per 1 g of cells, and sonicated (21 min total, 5 s on and 10 s off). Cell lysates were cleared by centrifugation (50,000 $\times$ g for 30 min at 4°C) and filtered through a 0.45 μM filter. For purification, lysates were applied to Ni-NTA resin (Thermo Scientific, Rockford, IL) that had been equilibrated with lysis buffer containing 20 mM imidazole. The protein was eluted with lysis buffer containing 400 mM imidazole. Protein was buffer exchanged to remove imidazole (50 mM Tris pH 8.0, 100 mM NaCl, and 2 mM βME) and the purification tag was cleaved. The protein was loaded on a HiPrep Q FF Anion Exchange Column (GE Healthcare Life Sciences, Marlborough, MA) and eluted by buffer with 300 mM NaCl. Recombinant ubiquitin variants were purified using the Ni-NTA affinity protocol described above, followed by size exclusion chromatography using a HiLoad Superdex 75 pg column (GE Healthcare Life Sciences, Marlborough, MA). The protein was eluted in a buffer containing (50 mM Tris pH 7.5, 150 mM NaCl, and 2 mM DTT). Ubiquitin-containing fractions were pooled and concentrated by centrifugation to 1 mM.

## Western blotting

Proteins in Laemmli buffer (for *E. coli* lysates) or urea sample buffer (for TCA-precipitated yeast proteins) containing 10% β-mercaptoethanol were resolved in 12–14% Bis-Tris PAGE gel by electrophoresis. Separated proteins were transferred onto PVDF membrane (0.2 μm, GE Healthcare Amersham) and immunoblotting was performed using the following primary antibodies: anti-ubiquitin (1:10,000; LifeSensors; MAb; clone VU-1), anti-K48 (1:10,000; Cell Signaling; RAb; clone D9D5), anti-K63 (1:4000; EMD Millipore; RAb; clone apu3), and anti-G6PDH (1:10,000; Sigma; RAb). Anti-mouse (IRDye 680RD-Goat anti-mouse) or anti-rabbit (IRDye 800CW-Goat anti-rabbit) secondary antibodies were purchased from LI-COR. Blots were visualized using Odyssey CLx (LI-COR Biosciences) and signal intensity was quantified using Image Studio Lite (LI-COR Biosciences).

## Mass spectrometry

SILAC-based mass spectrometry for quantitation and mapping of ubiquitin phosphorylation sites was performed as previously described (*Albuquerque et al., 2008*; *Lee et al., 2019*). Briefly, an equal amount of JMY1312 cells (labeled with either light or heavy Arg and Lys) expressing endogenous 3XFLAG-RPS31 and 3XFLAG-RPL40B were harvested from the mid-log phase and disrupted by bead beating using ice-cold lysis buffer (50 mM Tris-HCl, pH 7.5, 150 mM NaCl, 5 mM EDTA, 0.2% NP-40, 10 mM iodoacetamide, 1 mM 1,10-phenanthroline, 1$\times$ EDTA-free protease inhibitor cocktail [Roche], 1 mM phenylmethylsulfonyl fluoride, 20 μM MG132, 1$\times$ PhosStop [Roche], 10 mM NaF, 20 mM BGP, and 2 mM $Na_3VO_4$). Lysate was clarified by centrifugation at 21,000 $\times$ *g* for 10 min at 4°C and supernatant was transferred into a new tube and diluted with three-fold volume of ice-cold TBS (50 mM Tris-HCl, pH 7.5, 150 mM NaCl). 3XFLAG-ubiquitin in 12 mL diluted lysate was enriched by incubation with 50 μL of EZview anti-FLAG M2 resin slurry (Sigma) for 2 hr at 4°C with rotation. The resin was washed three times with cold TBS and incubated with 90 μL elution buffer (100 mM Tris-HCl, pH 8.0, 1% SDS) at 98°C for 5 min. The collected eluate was reduced with 10 mM DTT, alkylated with 20 mM iodoacetamide, and precipitated with 300 μL PPT solution (50% acetone, 49.9% ethanol, and 0.1% acetic acid). Light and heavy protein pellets were dissolved with Urea-Tris solution (8 M urea, 50 mM Tris-HCl, pH 8.0). Heavy and light samples were combined, diluted four-fold with water, and digested with 1 μg MS-grade trypsin (Gold, Promega) by overnight incubation at 37°C. Phosphopeptides were enriched by immobilized metal affinity chromatography (IMAC) using Fe(III)-

nitrilotriacetic acid resin as previously described (*MacGurn et al., 2011*) and dissolved in 0.1% trifluoroacetic acid. Peptides with K-ε-GlyGly remnant were isolated by immunoprecipitation as described in the PTMScan Ubiquitin Remnant Motif Kit (Cell Signaling Technologies) protocol. The GlyGly-peptide and phosphopeptide solutions were loaded on a capillary reverse-phase analytical column (360 µm O.D. × 100 µm I.D.) using a Dionex Ultimate 3000 nanoLC and autosampler and analyzed using a Q Exactive Plus mass spectrometer (Thermo Scientific). Data collected were searched using MaxQuant (ver. 1.6.5.0) and chromatograms were visualized using Skyline (ver. 20.1.0.31, MacCoss Lab).

### In vitro kinase activity

Kinase activity assays were performed in a reaction mixture containing 50 mM Tris (pH 7.4), 150 mM NaCl, 10 mM MgCl2, 0.1 mM ATP, 1 mM DTT, 0.5 µM ubiquitin, and 50 nM kinase. Reactions on linkage-specific ubiquitin polymers were carried out at 30°C for 30 min and quenched by adding an equal volume of Laemmli buffer with 10% β-mercaptoethanol and heating at 98°C for 10 min.

## Acknowledgements

We are very grateful to K Rose for technical advice and assistance with quantitative mass spectrometry analysis. We are also grateful to B Brasher for technical advice and recommending reagents. We also thank T Graham for critical reading of the manuscript. JMT was funded by NIH training grant T32 CA119925. This research was supported by NIH grant R21 AG053562 (to JAM), R01 GM118491 (to JAM), and R35 GM118089 (to WJC).

## Additional information

### Funding

| Funder | Grant reference number | Author |
| --- | --- | --- |
| National Institutes of Health | R21 AG053562 | Jason A MacGurn |
| National Institutes of Health | R01 GM118491 | Jason A MacGurn |
| National Institutes of Health | R35 GM118089 | Walter J Chazin |
| National Institutes of Health | T32 CA119925 | Jessica M Tumolo |

The funders had no role in study design, data collection and interpretation, or the decision to submit the work for publication.

### Author contributions

Nathaniel L Hepowit, Conceptualization, Data curation, Formal analysis, Investigation, Methodology, Writing - original draft, Writing - review and editing; Kevin N Pereira, Methodology; Jessica M Tumolo, Investigation; Walter J Chazin, Conceptualization, Resources; Jason A MacGurn, Conceptualization, Supervision, Funding acquisition, Investigation, Writing - original draft, Project administration, Writing - review and editing

### Author ORCIDs

Nathaniel L Hepowit (iD) https://orcid.org/0000-0002-7614-2756
Kevin N Pereira (iD) https://orcid.org/0000-0001-5377-397X
Walter J Chazin (iD) http://orcid.org/0000-0002-2180-0790
Jason A MacGurn (iD) https://orcid.org/0000-0001-5063-259X

### Decision letter and Author response

Decision letter https://doi.org/10.7554/eLife.58155.sa1
Author response https://doi.org/10.7554/eLife.58155.sa2

# Additional files

## Supplementary files

• Supplementary file 1. Corresponds to *Figure 2—figure supplement 1*. Yeast cells (NHY318 background) expressing either wildtype or S57D ubiquitin were cultured in heavy (H; expressing wildtype ubiquitin) or light (L, expressing S57D ubiquitin) SILAC media to the mid-log phase and treated with 1 mM $H_2O_2$ for 30 min before harvesting cells. Following cell lysis and digestion of lysates with trypsin for 24 hr, ubiquitin-remnant peptides were enriched (see Materials and methods) and analyzed by mass spectrometry. Three biological replicate experiments were analyzed. Since the peptide corresponding to K63-linked poly-ubiquitin also harbors the residue mutated in phosphomimetic (S57D) ubiquitin, K63 linkages are a blind spot for SILAC measurements in these experiments. 'n.d.' indicates not detected.

• Supplementary file 2. Corresponds to *Figure 3A* and *Figure 3—figure supplement 2*. For yeast Ser57 ubiquitin kinases, we analyzed consensus phosphorylation motifs as determined from a previous study based on in vitro activity analysis on peptide libraries (*Mok et al., 2010*). Values in parentheses are the quantified selectivity values, based on site preference of in vitro activity. Only amino acids selected at a position with a value >2 are shown.

• Transparent reporting form

## Data availability

All data generated or analyzed during this study are included in the manuscript and supporting files.

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
