## [Decision Letter]

**Acceptance summary:**

This work identifies candidate kinases for Ubiquitin in yeast, as well as their potential role in stress response poses foundations for further work in that undeniably interesting field.

**Decision letter after peer review:**

Thank you for submitting your article "Identification of ubiquitin Ser57 kinases regulating the oxidative stress response in yeast" for consideration by *eLife*. Your article has been reviewed by three peer reviewers, including Benoît Kornmann as the Reviewing Editor and Reviewer #1, and the evaluation has been overseen by a Reviewing Editor and David Ron as the Senior Editor.

The reviewers have discussed the reviews with one another and the Reviewing Editor has drafted this decision to help you prepare a revised submission.

This manuscript sets out to identify a role for ubiquitin phosphorylation and to identify the kinases necessary for it. The same group has previously shown that serine 57 phosphorylation can be detected in yeast cells. Here they generate strains expressing only phosphomimetic or phosphonull mutants and asses their phenotype in terms of Ubiquitin linkage alone and effect on cell physiology. Among other phenotypes, they find that a strain expressing a non phosphorylable ubiquitin likely fails to mount a response to low doses of H2O2, leading to a slightly increased sensitivity to this chemical. THey also find that treatment with H2O2 slightly increases the amount of phosphorylated Ubiquitin. They then go on to identify the kinases responsible for this phosphorylation using a screen in *E. coli*, which homes in two kinases, Vhs1 and Sks1.

They delete both kinase and show that this, to a large extent phenocopies the expression of a non-phosphorylable Ubiquitin, and that expression of a phosphomimetic rescued some of the phenotypes of the kinase deleted strain. They also show that overexpressing one of the kinase increases the amount of phosphorylation on ubiquitin.

Finally, they perform a similar screen using human kinases and human ubiquitin and identify a family of kinases that have the ability to phosphorylate ubiquitin in *E. coli*.

All three reviewers found the work of interest. Yet, because pSer57-Ubiquitin is so rare, they expressed concerns that the observed phosphorylation of Ubiquitin could be an epiphenomenon of little incidence to cell function.

First, the phenotypes of the alanine and aspartate mutants may be due to general effects on Ubiquitin rather than true phospho-Null and -mimetics effects. This concern is minimized by showing that the deletion and overexpression of the kinases phenocopy the ubiquitin mutants. Indeed, analysis of the ubiquitin mutant is only valid in the light of this phenocopy. Yet, because of its importance, this point can and should be pushed further. For instance, while the asp mutant is sensitive to hydroxyurea, the ala mutant behaves as a wildtype. This is at odds with the fact that the KO of each kinase individually increase HU resistance. In this case at least, the effect of deleting the kinase does not appear to involve a decrease in the level of ser57 phosphorylation. How can this be reconciled? Also, while you show that expressing the 57Asp bypasses the need for the kinase in the H2O2 sensitivity assay, is it also the case for the HU and tunicamicin resistance bestowed by the deletion of the kinases? Please find in the specific points a list of experiments required to better pinpoint the phenocopy that is so essential for the relevance of this study. Also, overexpression Vhs1 causes a slight canavanine resistance, reminiscent of canavanine resistance confered by s57d expression. Vhs1 overexpression should therefore not confer canavanine resistance if expressed in a s57a background. This is important to strengthen the phenocopy.

Second, while you show that both kinases can phosphorylate Ubiquitin in bacteria and in vitro, and that the overexpression of one of them increases the phosphorylation levels, you do not show how deletion of the kinase affect phosphorylation. This can and should be done, in particular to show if the observed increase in phosphorylation upon oxidative stress is mediated by these kinases.

Third, given the low abundance of pS57 ubiquitin, it is hard to conceive that this modification has an important effect on global chain linkage, unless this rare modification is applied to an equally rare set of substrates (like for instance PINK-1 mediated phosphorylation of ubiquitin is limited to the pool of ubiquitin that is on mitochondrial proteins). This should be better emphasized throughout the manuscript so as not to mislead readers into believing that a substantial fraction of ubiquitin is subjected to phosphorylation.

Fourth, in many cases, the experiments are not described to a sufficient amount of detail. For instance, vectors used herein are not described anywhere, nor is the way that all copies of ubiquitin have been replaced with mutant forms. The Supplementary file 2 is absent, so is Supplementary file 1. A much better method section is required to ensure the reproducibility of the experiments. Better descriptions of numbers pertaining to quantitative analysis, statistical test employed an p-value threshold, description of error bars (Stdev, SEM…) are also needed in figures and legends.

Essential revisions:

1) In Figure 1A and Figure 3—figure supplement 1, the authors test the effect of ubiquitin phospho-mutants and absence of kinases, respectively, on the ability of yeast cells to recover from acute heat stress. Firstly, it is puzzling though the experimental conditions are the same (39ᵒC for 18 hours and shifted back to 26ᵒC for recovery) in both cases, the wild-type strain is as good as dead in Figure 1A while it grows fine in Figure 3—figure supplement 1. Importantly, to validate the resistance phenotype of the S57D mutant, the authors should rather over-express the kinases and see that cells grow better in this condition compared to the wild-type and much better compared to the S57A mutant.

2) In Figure 1F, the authors employ anti-K48 ubiquitin and anti-K63 ubiquitin antibodies to show the specificity varies between S57A and the S57D mutant. The concern here is whether the serine mutants affect the binding of the antibodies. For instance, the epitope recognized by the anti-K63 ubiquitin antibody could involve the serine 57, however, when mutated to aspartate, the antibody can lose its ability to bind K63-linked ubiquitin. Is there a way to rule this out?

3) The authors show that S57D increase K48 but decrease K63-linkages whereas S57A decrease K48 but increase K63-linkages upon H2O2 treatment (Figure 1F). What about overexpression or deletion of Vhs1 and Sks1? Does absence of the kinases impact the mutual abundance of ubiquitin K48 and K63 linkages in vivo? Gly-Gly peptides analysis of the data in the experiment from Figure 2G might answer this.

4) Deletion of the kinases increase resistance to tunicamycin. However, expression of S57A does not. To strengthen the case of the phenocopy, it is important to check if kinases have ubiquitin-independet effects and how much of the phenocopy is actually wrought by independent mechanisms.

5) In general, the growth assays on tunicamycin, hydroxyurea or canavanin in Figure 1—figure supplement 1, Figure 3—figure supplement 2, Figure 3—figure supplement 3 and Figure3—figure supplement 4 should rather be moved to the main figures.

6) In Figure 4, human MARK kinases are found to trigger phosphorylation on UbS57 in vitro. It would be insightful to validate this finding in vivo and check whether phosphorylation of UbS57 also regulate the oxidative stress response in mammalian cells. I understand however that this might be take much longer to do than the timeframe which is allocated for revision. In this context, the authors may consider avoiding finishing the paper with these preliminary mammalian data and move them elsewhere in the manuscript. For instance, splitting data from Figure 2 in Figure 2 and Figure 3 and moving Figure 4C in the new Figure 2 (and Figure 4A and B in supplement) would save some space to end the paper with the current Figure 3 and its supplements.

Reviewer #1:

In this manuscript, Hepowit et al., unravel the yeast and the human kinases that phosphorylate ubiquitin at serine 57 and shed some light on the importance of this modification in conditions of stress. The authors back their claims with well designed and straightforward experiments. Phosphorylation of ubiquitin plays important roles, the extent of which is not yet fully understood. The findings in this paper are intriguing, important and pave the way to further research.

Essential revisions:

1) In Figure 1A and Figure 3—figure supplement 1, the authors test the effect of ubiquitin phospho-mutants and absence of kinases, respectively, on the ability of yeast cells to recover from acute heat stress. Firstly, it is puzzling though the experimental conditions are the same (39ᵒC for 18 hours and shifted back to 26ᵒC for recovery) in both cases, the wild-type strain is as good as dead in Figure 1A while it grows fine in Figure 3—figure supplement 1. Importantly, to validate the resistance phenotype of the S57D mutant, the authors should rather over-express the kinases and see that cells grow better in this condition compared to the wild-type and much better compared to the S57A mutant.

2) In Figure 1F, the authors employ anti-K48 ubiquitin and anti-K63 ubiquitin antibodies to show the specificity varies between S57A and the S57D mutant. The concern here is whether the serine mutants affect the binding of the antibodies. For instance, the epitope recognized by the anti-K63 ubiquitin antibody could involve the serine 57, however, when mutated to aspartate, the antibody can lose its ability to bind K63 ubiquitin. Is there a way to rule this out?

Reviewer #3:

Modifying the modifier is a principle that took all its meaning few years ago when Ubiquitin was found to be phosphorylated on Serine 65 to regulate mitophagy in mammalian cells through a crosstalk between the ubiquitin ligase Parkin and the mitochondrial kinase PINK1. Since then, MacGurn and colleagues have shown that Ubiquitin also undergoes phosphorylation in yeast on Serine 57 but kinases involved in this phosphorylation remain to be identified. In this short report, the same lab now finds that phosphorylation of UbSer57 is required for heat shock and oxidative stress responses in yeast and identifies two kinases of the Snf1-related family that mediate this phosphorylation. The experimental design of this study is very solid, the manuscript is very well written and the identification of kinases promoting ubiquitin phosphorylation in yeast is of high enough significance to warrant rapid publication in *eLife*. Besides aspects that are clearly out of scope for this short report (identification of ligases or substrates), I must admit that my suggestions for improvement of this study are only cosmetic.

Essential revisions:

- The authors show that S57D increase K48 but decrease K63-linkages whereas S57A decrease K48 but increase K63-linkages upon H2O2 treatment (Figure 1F). What about overexpression or deletion of Vhs1 and Sks1? Does absence of the kinases impact the mutual abundance of ubiquitin K48 and K63 linkages in vivo? Any Gly-Gly peptides detected in the experiment from Figure 2G?

- Deletion of the kinases mimic the growth effects of S57A on tunicamycin and hydroxyurea. It would be nice to check whether UbS57D can bypass the absence of the kinases in these conditions even if this has been evaluated for the responses to oxidative stress in Figure 3.

- In general, the growth assays on tunicamycin, hydroxyurea or canavanin in Figure 1—figure supplement 1, Figure 3—figure supplement 2, Figure 3—figure supplement 3 and Figure3—figure supplement 4 should rather be moved to the main figures.

- In Figure 4, human MARK kinases are found to trigger phosphorylation on UbS57 in vitro. It would be insightful to validate this finding in vivo and check whether phosphorylation of UbS57 also regulate the oxidative stress response in mammalian cells. I understand however that this might be take much longer to do than the timeframe which is allocated for revision. In this context, the authors may consider avoiding finishing the paper with these preliminary mammalian data and move them elsewhere in the manuscript. For instance, splitting data from Figure 2 in Figure 2 and Figure 3 and moving Figure 4C in the new Figure 2 (and Figure 4A and B in supplement) would save some space to end the paper with the current Figure 3 and its supplements.

Reviewer #4:

This manuscript by Hepowitt et al., focusses on the mechanism and phenotypic consequences of ubiquitin post-translational modification (PTM). Specifically, one of the first identified modifications – phosphorylation of Ser57. This loosely follows on from previous work conducted in the MacGurn lab in which they identified ubiquitin Ser57 phosphorylation (Ser57P) as a regulator of endocytic trafficking and ubiquitin turnover (Lee et al., 2017).

The current manuscript begins by presenting a number of varied phenotypes associated with yeast solely expressing Ser57Asp (a phosphomimetic) and Ser57Ala (non phosphorylatable) ubiquitin mutants (presumably from an endogenous ubiquitin loci). Yeast expressing Ser57Asp have a conspicuous heat tolerance phenotype (Figure 1A-C) and an increased sensitivity to hydroxyurea relative to wildtype yeast (Figure S1A). Curiously, the opposite trend is not observed with Ser57Ala ubiquitin. Conversely, Ser57Asp and Ser57Ala ubiquitin mutants do possess contrasting phenotypes, with respect to wildtype yeast, in terms of tunicamycin sensitivity (which increases in Ser57Asp and decreases in Ser57Ala expressing strains, respectively) (Figure S1A).

The authors then move towards the principal thesis of their manuscript; that modification of Ser57 is important for the proper regulation of the yeast oxidative stress response. In support of this tenet Hepowitt et al. present the somewhat contradictory observations that Ser57Ala ubiquitin permits more growth in the presence of H2O2 (Figure 1D) but reduced viability in response to a shorter H2O2 stress (Figure 1E). Ser57Asp mutation does not perturb the yeast in these assays, relative to wildtype strains. The authors suggest that these differences in response to oxidative stress may play out at the level of different polyubiquitin linkage types favoured by the different amino acids at residue 57 (as a proxy for different phosphorylation statuses). This is supported by the observation that Ser57Asp yeast have significantly elevated levels of poly-K48-linked ubiquitin and decreased levels of poly-K63-linked in response to H2O2 exposure and also seemingly under basal/0 mM H2O2 conditions, relative to wildtype (Figure 1F). The reciprocal phenotypes were also observed for Ser57Ala yeast. This observation, in turn, stimulates the authors to speculate that oxidative stress may induce the phosphorylation of Ser57 which subsequently favours poly-K48-linked ubiquitination and thus proteasomal degradation of the increased burden of oxidised/damaged protein. Accordingly, reduced carbonylated proteins are observed in Ser57Asp strains (Figure 1G). However, given the contrast between Ser57Asp and wildtype cells in terms of ubiquitin linkage both {plus minus} H2O2 (Figure 1F) the above hypothesis is conceptually difficult to marry with the observation of seeming parity between wildtype and Ser57Asp expressing yeast in terms of oxidative stress growth response and post-stress viability (Figure 1D-E) and with the observation that wildtype cells both increase K48 and K63-linked polyubiquitin in response to oxidative stress (Figure 1F).

An important experiment, and the first time the authors address the phosphorylation of Ser57 directly in this manuscript, is presented in Figure 1F. Based on this SILAC experiment it appears that oxidative stress (H2O2) can induce up to around a 1.7-fold increase in the amount of Ser57P ubiquitin. The authors then go on to demonstrate quite nicely that two SnfI-related Ser/Thr kinases VhsI and SksI can phosphorylate ubiquitin Ser57 in vitro, in an *E. coli* co-expression system and when SksI is overexpressed in yeast (the latter causing up to a 4-fold increase in Ser57P levels, Figure 1G). Crucially, however, the authors do not go on to show that endogenous VhsI and/or SksI directly influence the modification of endogenous ubiquitin at Ser57 in yeast and, by extension, they also do not show that these kinases regulate Ser57P in an oxidative stress dependent manner. That is to say, in order to substantiate their claim that these kinases regulate yeast oxidative stress response by phosphorylating Ser57 a similar experiment to Figure 1H, comparing the relative abundance of Ser57P peptides in wildtype and Δsks1Δvhs1 cells {plus minus} H2O2, should have been undertaken.

Instead, the authors compare kinase knockout phenotypes to the previous Ser51Asp and Ser51Ala phenotypes. Unfortunately, even these are performed in a rather haphazard fashion. For example, although Ser57Asp was shown to produce a robust heat tolerance phenotype and also a tunicamycin sensitivity (Figure 1A-C and S1A), in Figure S3C wildtype cells were only compared in this regard to Δsks1Δvhs1 and not also to SksI/VhsI over-expression (or to Ppz1/2 knockout, the phosphatase identified by this group in Lee at al., (2017) to be responsible for Ser57P dephosphorylation and surprisingly not mentioned or utilised in this paper). With respect to assaying the role of these kinases (and by extension the function of Ser57P) in an oxidative stress response the authors only examine the growth response to H2O2 (Figure 3). The presence of both kinases appears necessary for growth arrest, aligning with the lack of growth arrest previously observed in the Ser57Ala context. In order to more definitively demonstrate that this phenotype is due to an inability to phosphorylate ubiquitin's Ser57 the analysis in Figure 3 should have been extended. That is to say, the phenotypic rescue of Δsks1Δvhs1 by overexpression of SksI+VhsI should have been tested in different ubiquitin backgrounds. If the rescue is dependent on SksI+VhsI mediated phosphorylation of Ser57 then the growth Δsks1Δvhs1 phenotype should not be rescuable in the background of Ser57Ala mutant ubiquitin.

In summary, the authors do not adequately demonstrate that endogenous SksI and VhsI kinases are responsible for phosphorylation of endogenous ubiquitin at Ser57 either in the absence or presence of an oxidative stressor. As such, the claim that these kinases and by extension ubiquitin phosphorylation at Ser57 is important in the yeast oxidative stress response is poorly supported. The authors imply, largely based on replacement of the entire ubiquitin pool with either Ser57Ala or Ser57Asp mutants, that in response to oxidative stress SksI/VhsI phosphorylates Ser57 (with a strong preference for polyubiquitin, Figure 2E-F). This, in turn, results in an increased propensity for poly-K48-linkages and thus increased proteasomal clearance of oxidised or damaged proteins and by some mechanism an oxidative stress induced growth retardation (Figure 1). However, what is conspicuous by its absence is a lack of discussion regarding the stoichiometry of this modification. It was previously estimated by this group, using mass spectrometry, that in yeast lacking the Ser57 phosphatases (Ppz1 and Ppz2), which possessed a ca. 3-fold elevation in Ser57P modification, that the amount of ubiquitin modified at Ser57 was still less than 0.05% of the entire ubiquitin population (Lee at al., 2017). In this manuscript the authors find that oxidative stress causes less than a 2-fold upregulation in Ser57P levels, this would therefore represent less than 0.03% of the ubiquitin population being phosphorylated at Ser57 even after H2O2 application. The inferences that can therefore be drawn from replacing all of the endogenous ubiquitin with Ser57Ala and Ser57Asp (which may themselves have perturbing effects on the function and biochemistry of this highly conserved protein and may not be particularly representative of unmodified and phosphorylated Ser57, respectively) are extremely limited. The mechanism by which such a minor fraction of the ubiquitin pool could affect the oxidative stress response is a conundrum which is not at all addressed by Hepowit et al.

[Editors' note: further revisions were suggested prior to acceptance, as described below.]

Thank you for submitting your article "Identification of ubiquitin Ser57 kinases regulating the oxidative stress response in yeast" for consideration by *eLife*. Your article has been reviewed by three peer reviewers, including Benoît Kornmann as the Reviewing Editor and Reviewer #1, and the evaluation has been overseen by a Reviewing Editor and David Ron as the Senior Editor.

The reviewers have discussed the reviews with one another and the Reviewing Editor has drafted this decision to help you prepare a revised submission.

You will be glad to learn that, despite some disagreement among reviewers, the consensus was that your story might deserve publication.

While all three reviewers acknowledge the potential importance of your work, they also acknowledge that you have not formally proven that VHS1 and SKS1 are kinases that phosphorylate ubiquitin in vivo and that this is important for stress response. What you have shown is that these two kinases are sufficient for phosphorylation in bacteria, but that they are not necessary for phosphorylation in vivo.

Your interpretation that Vhs1 and Sks1 are necessary for the phosphorylation of a defined subset of ubiquitin molecules (as is the case for PINK1) is plausible, but far from being the only interpretation.

Indeed, many of your experiments that assess fitness of strains with ubiquitin variants and kinase KO or overexpression hinge on the prerequisite that phosphomimic or phosphonull mutants behave as expected and do not affect any other aspect of ubiquitin biology, a point that you cannot ascertain.

Moreover the phosphomimic and phosphonull mutants are bound to have effects beyond what can be attributed to Vhs1 and Sks1, the mimics because they affect 100% of ubiquitins, while phospho-ubiquitins at maximum represent a small fraction, and the null mutants because they will affect non-Vhs1/Sks1-mediated phosphorylation, which is the majority of phosphorylation. Moreover, simply missing a hydroxyl group on the surface of ubiquitin might alter its properties, irrespective of phosphorylation.

So an equally plausible explanation is that ubiquitin biology is altered by your mutations in way which happens to phenocopy the kinases KO and overexpression, respectively, at least at the level of general fitness and stress survival.

The disagreement between the reviewers hinged upon allowing you the benefit of the doubt, or considering that the doubt was too great to constitute a publishable story. The consensus reached was that your paper deserves publication, provided that you make the following amendments to it:

1) The new proteomics results, as disappointing as you might find them, are an important part of the story and should constitute part of a main figure and not be buried in a Supplementary file. They should be the basis of an important discussion point (see below).

2) Instead of a result and Discussion section, the paper should have a stand-alone Discussion section, where these concerns should be thoroughly put forward.

3) In this discussion you should explicitly state that you are asking readers to accept the proposition that the phenotype in question arises from a small Sks1/Vhs1-dependent pool of Ub_pS57, which is undetectable on the background levels of Ub_pS57, which are themselves a very small fraction of Ub.

All reviewers sincerely hope that your views will be vindicated in due course, and that you will enjoy the satisfaction of knowing that you persevered for good reason in face of considerable skepticism, but in the meantime, it is important for all interested parties, to lay bare the limitations of your current study.

The individual reviews are appended for your perusal.

Reviewer #1:

The revised manuscript submitted here by Hepowit et al. addresses many of the concerns raised by the reviewers. In the first round of review and the manuscript presented here is therefore much improved. Still, the new data are not necessarily aligned with the authors’ original model. Some clarifications are therefore required.

1) Importantly, the model is that two kinases phosphorylate ubiquitin under stress. When directly tested in strains deficient for these two kinases, it appears that the kinases are not necessary for stress-induced phosphorylation of ubiquitin. This is an important result that deserves better than being mentioned and kept for a Supplementary file 3. Moreover, it is unclear what a SILAC ratio of 1 actually means since in all other figures, SILAC ratios are presented as log2. Here it appears to be on a linear scale (with 1 meaning 'no change'). This should be clarified and the data from Supplementary file 1 presented in a more visible fashion in the main figure. The fact that neither Sks1 nor Vph1 are required for stress induced phosphorylation of ubiquitin is too important to the comprehension of the implications of this paper.

2) It was shown in the first version of the paper that individual mutation of the kinases led to HU resistance (previous Figure 3—figure supplement 4). This was at odd with the finding that the expression of a non-phosphorylatable ubiquitin did not cause the same phenotype, indicating that both kinases exert their function on HU resistance through ubiquitin-independent mechanisms. These data have now disappeared from the manuscript. Although the interpretation of these results is not straightforward, it is something that readers will want to see, nonetheless.

Reviewer #2:

Taken together, the authors have made a good job in revising the manuscript and in taking reviewers comments into consideration. It is clearly disappointing that the combined absence of Vhs1 and Sks1 fails to induce detectable decrease in phosphorylation of UbS57. Yet, this can be explained by a possible specificity of Vhs1 and Sks1 toward limited ubiquitinated substrates and by the fact that other kinases, distinct form Vhs1 and Sks1, have the capacity to phosphorylate UbS57. While the authors emphasize these possibilities, they also provide several data that support this interpretation. As shown in the initial version of the manuscript, Gin4 and Kcc4 are two other kinases that can trigger UbS57 phosphorylation. Most importantly, new data from the revised version now support a possible specificity of Vhs1-mediated phosphorylation of UbS57 in protein misfolding stress (new Figure 4C) and of Sks1-mediated phosphorylation of UbS57 in DNA damage stress (new Figure 4E). Notably, Figures 3E and 3F also demonstrate that the overexpression of Sks1 and of Vhs1 both stimulate the phosphorylation of endogenous Ubiquitin at S57.

In the eyes of this reviewer, the revised manuscript is really improved and the new data appropriately support the claims of the authors.

Reviewer #3:

The claim for discovery here is that phosphorylation of Ub on serine 57 affects stress responsiveness of yeast.

The key evidence in support of this conclusion is the concordance in phenotype between a replacement allele that renders Ser57 unphosphorylateable (S57A) and inactivating mutations of two kinases demonstrated in a heterologous system to be capable of phosphorylating Ub_S57 [the authors also provide information on the phenotype of an Ub_S57D 'phosphomimetic' mutation and gain of function features of the kinases, but these give rise to a less clear cut picture and are thus ignored for the purpose of this review]

How credible is this interpretation?

The stoichiometry of S57 phosphorylation is exceedingly low, which is to say that in stressed cells most of the Ub remains unphosphorylated. To account for the phenotypic consequences of kinase inactivation and/or Ub_S57A one would have to conjure a scenario whereby phosphorylation affects a small pool of Ub that is critical to the stress adaptation. Given the potential precedent set by PINK1, this was a speculation we as reviewers were willing to sanction when we invited a revised version of the paper without insisting that evidence for the existence of such a pool be provided.

In the revised version of the manuscript we now learn that the kinases in question, Sks1 and Vhs1 are not essential for the stress-induced Ub phosphorylation – other kinases can compensate for this biochemical event, but not, apparently, for its phenotypic consequences. Thus, to uphold the paper's key conclusion we must now be willing to take an additional leap of faith, namely that the already small pool of Ub_pS57 is itself comprised of two sub-pools, one that is Sks1 and Vhs1-dependent – and important to the phenotype arising from the Ub_S57A mutation (and/or the inactivation of the kinases) – and a second pool that is irrelevant to this stress response.

As reviewers we need to ask ourselves if we are willing to endorse the publication of a paper that makes the aforementioned claim, supports it with the data provided here and then qualifies the conclusion with these key caveats relating to the unproven existence of a critical small pool of Sks1 and Vhs1-dependent Ub_pS57.

---

## [Author Response]

All three reviewers found the work of interest. Yet, because pSer57-Ubiquitin is so rare, they expressed concerns that the observed phosphorylation of Ubiquitin could be an epiphenomenon of little incidence to cell function.

We are extremely grateful for the reviewers’ interest in the manuscript, and we agree with the concern that ubiquitin phosphorylation could be an epiphenomenon. However, we think these concerns are mitigated by the following:

- Several new lines of genetic evidence have been added to the revised manuscript which expand upon the genetic relationship between the Ser57 ubiquitin kinases that we identified and the phosphorylation of ubiquitin at the Ser57 position. The details of these new experiments are outlined in the following point-by-point response to reviewer comments.

- We think the data strongly indicate that Ser57 phosphorylation of ubiquitin plays a physiological role in cell stress responses. Biochemical data indicates that Ser57 phosphorylated ubiquitin is produced in response to conditions of oxidative stress, and genetic data indicates Ser57 phosphorylated ubiquitin is a requirement for the oxidative stress response. Our data also implicates a functional role for Ser57 phosphorylation in other proteotoxic stress responses.

- Although we cannot exclude the possibility that ubiquitin phosphorylation is an epiphenomenon, the stress tolerance phenotypes are real and therefore may have synthetic value. For example, mechanisms that promote cellular tolerance to proteotoxic stresses may ultimately have therapeutic value for diseases associated with protein misfolding. Additionally, yeast variants that exhibit thermal tolerance may have industrial applications. Therefore, the results reported in this study have potential impact beyond the understanding of cellular responses.

Overall, we believe the new experiments and revisions incorporated have resulted in a stronger manuscript that will be of significant interest to many labs in different fields – and we hope the editors and reviewers agree.

First, the phenotypes of the alanine and aspartate mutants may be due to general effects on Ubiquitin rather than true phospho-Null and -mimetics effects. This concern is minimized by showing that the deletion and overexpression of the kinases phenocopy the ubiquitin mutants. Indeed, analysis of the ubiquitin mutant is only valid in the light of this phenocopy. Yet, because of its importance, this point can and should be pushed further. For instance, while the asp mutant is sensitive to hydroxyurea, the ala mutant behaves as a wildtype. This is at odds with the fact that the KO of each kinase individually increase HU resistance. In this case at least, the effect of deleting the kinase does not appear to involve a decrease in the level of ser57 phosphorylation. How can this be reconciled? Also, while you show that expressing the 57Asp bypasses the need for the kinase in the H2O2 sensitivity assay, is it also the case for the HU and tunicamicin resistance bestowed by the deletion of the kinases? Please find in the specific points a list of experiments required to better pinpoint the phenocopy that is so essential for the relevance of this study. Also, overexpression Vhs1 causes a slight canavanine resistance, reminiscent of canavanine resistance confered by s57d expression. Vhs1 overexpression should therefore not confer canavanine resistance if expressed in a s57a background. This is important to strengthen the phenocopy.

These are excellent points raised by reviewers. In the revised manuscript, we have expanded our genetic analysis, which has revealed additional genetic interactions between the Ser57 ubiquitin kinases and the Ser57 position of ubiquitin. Specifically, we have included the following new experiments in Figure 4:

- Overexpression of VHS1 phenocopies the canavanine resistance phenotype observed with expression of S57D ubiquitin. This phenotype requires catalytic activity of Vhs1, and it is suppressed by expression of S57A ubiquitin.

- Overexpression of VHS1 phenocopies the thialysine resistance phenotype observed with expression of S57D ubiquitin. This phenotype requires catalytic activity of Vhs1, and it is suppressed by expression of S57A ubiquitin.

- Overexpression of SKS1 phenocopies the hydroxyurea sensitivity phenotype observed with expression of S57D ubiquitin. This phenotype is suppressed by expression of S57A ubiquitin.

All of these findings indicate that overexpression of SKS1 and VHS1 drive production of Ser57 phosphorylated ubiquitin (as shown in Figure 3E-F), which is required for the observed phenotypes. Combined with our original genetic analysis of these kinases in the context of oxidative stress, we think these results strengthen the main conclusions of the paper.

(Please note that we have also decided to remove our analysis of growth in the presence of tunicamycin. The tunicamycin resistance phenotype of *Δsks1Δvhs1* double mutant cells is subtle and will require additional genetic analysis in follow-up studies. However, we do not believe this exclusion changes the main conclusions or significance of the manuscript.)

Second, while you show that both kinases can phosphorylate Ubiquitin in bacteria and in vitro, and that the overexpression of one of them increases the phosphorylation levels, you do not show how deletion of the kinase affect phosphorylation. This can and should be done, in particular to show if the observed increase in phosphorylation upon oxidative stress is mediated by these kinases.

This experiment was the major bottleneck to our resubmission, since it required us to generate new strains and due to significant delays at our proteomics core facility. We performed the requested experiment, and the data are now presented in Figure 4, Supplementary file 3. In this SILAC experiment, we compared wildtype and *Δsks1Δvhs1* double mutant cells in regular growth conditions (Experiment #1) and following a 30 minute exposure to H2O2 (Experiment #2) and in neither case did we observe significant change in the amount of Ser57 phospho-ubiquitin. This result indicates that absence of Sks1 and Vhs1 does not affect global levels of Ser57 phosphorylated ubiquitin following acute exposure to H2O2. However, it does not exclude the possibility that these kinases may participate in the response to prolonged oxidative stress, and it does not account for localized production of phosphorylated ubiquitin. Redundancy with other kinases (Gin4, Kcc4, or other as-yet unidentified Ser57 ubiquitin kinases) may also be a factor – although redundancy is not consistent with the genetic requirement of *SKS1* and *VHS1* for growth arrest during oxidative stress. We have tried to objectively interpret the data (Results section) and we hope reviewers and editors will appreciate our effort despite the outcome.

Third, given the low abundance of pS57 ubiquitin, it is hard to conceive that this modification has an important effect on global chain linkage, unless this rare modification is applied to an equally rare set of substrates (like for instance PINK-1 mediated phosphorylation of ubiquitin is limited to the pool of ubiquitin that is on mitochondrial proteins). This should be better emphasized throughout the manuscript so as not to mislead readers into believing that a substantial fraction of ubiquitin is subjected to phosphorylation.

We agree that in physiological circumstances the production of phospho-ubiquitin is highly localized – as is the case for Pink1-mediated production of Ser65 phosphorylated ubiquitin. Indeed, this interpretation is consistent with many of our observations presented throughout the manuscript, and this is also the reason why genetics has been a powerful tool for accessing the biology of Ser57 phosphorylated ubiquitin. To emphasize this point, we have made the following changes and additions to the text:

- Results section: By way of introducing the yeast ubiquitin replacement strains used in this study (S57A and S57D) we emphasize that this strategy inherently over-estimates the stoichiometry of ubiquitin phosphorylation, and thus the phenotypes likely exaggerate effects that are likely localized and transient in a physiological context.

- Results section: In discussing the impact S57A and S57D have on linkage types used in conjugation, we re-emphasize the point that complete ubiquitin replacement in the strains used exaggerates the effects expected at physiological phosphorylation levels. We also link this to discussion of our result that Ser57 phosphorylation is induced by oxidative stress, but the stoichiometry is still sufficiently low that the impact is likely highly localized.

- Results section: In this concluding paragraph, we emphasize the low stoichiometry of ubiquitin phosphorylation in a physiological context. We also compare Ser57 kinases to Pink1, speculating that activity of these kinases on localized pools of ubiquitin likely drives the biology that is underscored by the genetics.

We think these changes capture the spirit of the reviewer/editorial critique. In doing so, we have more clearly articulated the limitations associated with the data we present, and we believe this strengthens the manuscript as a whole.

Fourth, in many cases, the experiments are not described to a sufficient amount of detail. For instance, vectors used herein are not described anywhere, nor is the way that all copies of ubiquitin have been replaced with mutant forms. The Supplementary file 2 is absent, so is Supplementary file 1. A much better method section is required to ensure the reproducibility of the experiments. Better descriptions of numbers pertaining to quantitative analysis, statistical test employed an p-value threshold, description of error bars (Stdev, SEM…) are also needed in figures and legends.

We apologize for these oversights, which have been addressed in the revised manuscript. Here is a list of specific changes and additions to address this concern:

- All strains and plasmids used in this study are now described in a Key Resources Table.

- We have also provided in the Materials and methods section a better description of our ubiquitin replacement yeast strains and how those were generated.

- In Figure Legends, we now provide a thorough description of statistical tests, error bars, significance thresholds, and numbers of biological replicates for all relevant experiments depicted in main and supplemental figures.

- We have included source data files corresponding to each figure, which includes all numerical data and statistical analysis that was used to generate all graphs presented in this study.

We have expanded the experimental details provided in our Materials and methods section to ensure reproducibility of the experiments.

Essential revisions:1) In Figure 1A and Figure 3—figure supplement 1, the authors test the effect of ubiquitin phospho-mutants and absence of kinases, respectively, on the ability of yeast cells to recover from acute heat stress. Firstly, it is puzzling though the experimental conditions are the same (39ᵒC for 18 hours and shifted back to 26ᵒC for recovery) in both cases, the wild-type strain is as good as dead in Figure 1A while it grows fine in Figure 3—figure supplement 1.

This disparity is due to differences in strain background between these two experiments. For cells expressing only WT, S57A, or S57D ubiquitin, we have used the SUB280 background, which was designed to express ubiquitin from a single source and is therefore used in experiments for phenotypic analysis of strains expressing S57A and S57D ubiquitin. (Figure 4—figure supplement 2) was performed in the SEY6210 strain background, which has greater thermal tolerance than SUB280. In the revised manuscript, we have added text in the Materials and methods to clarify why SUB280 and SEY6210 background were used in different experiments.

Importantly, to validate the resistance phenotype of the S57D mutant, the authors should rather over-express the kinases and see that cells grow better in this condition compared to the wild-type and much better compared to the S57A mutant.

These are good suggestions raised by reviewers. Our analysis did not uncover any thermal tolerance phenotypes associated with the deletion or overexpression of Ser57 ubiquitin kinases identified in this study (Figure 4—figure supplement 1, Figure 4—figure supplement 2). However, this does not exclude a role for Ser57 phosphorylation in the heat stress response, and we hypothesize there may be additional (and as yet unidentified) Ser57 ubiquitin kinases that function specifically in the heat stress response.

We have also included new data revealing that yeast cells expressing S57A ubiquitin exhibit thermal sensitivity (Figure 1E and Figure 1—figure supplement 1). This provides additional evidence of a role for Ser57 phosphorylated ubiquitin in promoting thermal tolerance, which we think strengthens the revised manuscript.

2) In Figure 1F, the authors employ anti-K48 ubiquitin and anti-K63 ubiquitin antibodies to show the specificity varies between S57A and the S57D mutant. The concern here is whether the serine mutants affect the binding of the antibodies. For instance, the epitope recognized by the anti-K63 ubiquitin antibody could involve the serine 57, however, when mutated to aspartate, the antibody can lose its ability to bind K63-linked ubiquitin. Is there a way to rule this out?

This is an excellent point raised by reviewers. Signal from blotting with the anti-K63 antibody is typically very weak, but we actually detect a baseline signal for K63 linkage detection that declines with oxidative stress (Figure 2C). We have also now quantified this over triplicate experiments (Figure 2D-E). These data suggest that S57D ubiquitin is recognized by the anti-K63 antibody but the signal decreases with oxidative stress.

To validate the findings of Figure 2C-E using methods that do not rely on linkage-specific antibodies, we turned to SILAC-MS combined with enrichment of ubiquitin-remnant (di-Gly) peptides comparing cells expressing wildtype (heavy-labelled) and S57D (light-labelled) ubiquitin. In this analysis, we resolved linkage-specific peptides corresponding to K6-, K33- and K48-linked polymers (Figure 2, Supplementary file 1). Consistent with our blotting analysis, SILAC quantification revealed a 39% increase in K48-linked polymers associated with S57D ubiquitin. Importantly, K63-linked polymers are a blind spot of this analysis, since the Ser57Asp mutation occurs on the same peptide that harbors K63. (Thus, these peptides differ between the two samples and cannot be quantified by SILAC.) Nevertheless, these results are consistent with immunoblotting results presented in Figure 2, and they provide additional quantitative analysis of other linkage-types not amenable to immunoblot analysis (e.g., K6- and K33linkages). We believe the addition of this new experimental data addresses the reviewers concerns and strengthens the manuscript.

3) The authors show that S57D increase K48 but decrease K63-linkages whereas S57A decrease K48 but increase K63-linkages upon H2O2 treatment (Figure 1F). What about overexpression or deletion of Vhs1 and Sks1? Does absence of the kinases impact the mutual abundance of ubiquitin K48 and K63 linkages in vivo? Gly-Gly peptides analysis of the data in the experiment from Figure 2G might answer this.

This is an excellent idea – and we went back over the data for Figures 3E-F (formerly Figure 2G). Although some linkage-specific diGly peptides were resolved in this analysis, there are no consistent trends for any peptides other than the Ser57 phosphopeptide in these experiments. This finding suggests that production of Ser57 phospho-ubiquitin, at least at the levels attained in these experiments, is not sufficient to substantially affect global ubiquitin linkage patterns. The source data for these experiments is now provided in Figure 3—source data 1.

4) Deletion of the kinases increase resistance to tunicamycin. However, expression of S57A does not. To strengthen the case of the phenocopy, it is important to check if kinases have ubiquitin-independet effects and how much of the phenocopy is actually wrought by independent mechanisms.

Actually, the initial submission included data revealing that kinase deletion and S57A expression both resulted in slight resistance to tunicamycin, but because this phenotype was subtle in both cases we decided to remove the tunicamycin results from this study in favor of stronger phenotypes. Instead, we have included new data for canavanine, thialysine, and hydroxyurea phenotypes that link the kinases to ubiquitin-dependent effects. We think this strengthens the revised manuscript.

5) In general, the growth assays on tunicamycin, hydroxyurea or canavanin in Figure 1—figure supplement 1, Figure 3—figure supplement 2, Figure 3—figure supplement 3 and Figure3—figure supplement 4 should rather be moved to the main figures.

This is an excellent suggestion – and we have moved more phenotypic data to the main figures (Figure 4) in the revised manuscript.

6) In Figure 4, human MARK kinases are found to trigger phosphorylation on UbS57 in vitro. It would be insightful to validate this finding in vivo and check whether phosphorylation of UbS57 also regulate the oxidative stress response in mammalian cells. I understand however that this might be take much longer to do than the timeframe which is allocated for revision. In this context, the authors may consider avoiding finishing the paper with these preliminary mammalian data and move them elsewhere in the manuscript. For instance, splitting data from Figure 2 in Figure 2 and Figure 3 and moving Figure 4C in the new Figure 2 (and Figure 4A and B in supplement) would save some space to end the paper with the current Figure 3 and its supplements.

This is an excellent suggestion, and we have re-structured the revised manuscript accordingly.

Reviewer #4:This manuscript by Hepowitt et al., focusses on the mechanism and phenotypic consequences of ubiquitin post-translational modification (PTM). Specifically, one of the first identified modifications – phosphorylation of Ser57. This loosely follows on from previous work conducted in the MacGurn lab in which they identified ubiquitin Ser57 phosphorylation (Ser57P) as a regulator of endocytic trafficking and ubiquitin turnover (Lee et al., 2017).The current manuscript begins by presenting a number of varied phenotypes associated with yeast solely expressing Ser57Asp (a phosphomimetic) and Ser57Ala (non phosphorylatable) ubiquitin mutants (presumably from an endogenous ubiquitin loci). Yeast expressing Ser57Asp have a conspicuous heat tolerance phenotype (Figure 1A-C) and an increased sensitivity to hydroxyurea relative to wildtype yeast (Figure S1A). Curiously, the opposite trend is not observed with Ser57Ala ubiquitin. Conversely, Ser57Asp and Ser57Ala ubiquitin mutants do possess contrasting phenotypes, with respect to wildtype yeast, in terms of tunicamycin sensitivity (which increases in Ser57Asp and decreases in Ser57Ala expressing strains, respectively) (Figure S1A).

These are excellent observations. We note that we have included additional data in the revised manuscript that reveal a heat sensitivity phenotype of yeast expression S57A ubiquitin (Figure 1E and Figure 1—figure supplement 1). We have also decided to remove our analysis of growth in the presence of tunicamycin. The tunicamycin resistance phenotype of *Δsks1Δvhs1* double mutant cells is subtle and will require additional genetic analysis in follow-up studies. However, we do not believe this exclusion changes the main conclusions or significance of the manuscript.

The authors then move towards the principal thesis of their manuscript; that modification of Ser57 is important for the proper regulation of the yeast oxidative stress response. […] However, given the contrast between Ser57Asp and wildtype cells in terms of ubiquitin linkage both {plus minus} H2O2 (Figure 1F) the above hypothesis is conceptually difficult to marry with the observation of seeming parity between wildtype and Ser57Asp expressing yeast in terms of oxidative stress growth response and post-stress viability (Figure 1D-E) and with the observation that wildtype cells both increase K48 and K63-linked polyubiquitin in response to oxidative stress (Figure 1F).

In consideration of these excellent points raised by reviewer #4, we have more rigorously addressed and quantified the linkage-type affects associated with expression of S57A and S57D ubiquitin in yeast (Figure 2C-E and Figure 2—figure supplement 2 and Supplementary file 1). While the phenotypic experiments (Figure 2A-B) do not reveal any difference between yeast cells expressing wildtype and those expression S57D ubiquitin variants, it is notable that no loss of viability is detected in wildtype cells (Figure 2B) so we cannot conclude phenotypic parity between wildtype and S57D expressing yeast. Furthermore, while there does appear to be an increase in K63-linked polyubiquitin in response to oxidative stress, quantification of our results does not reveal this to be statistically significant. This may reflect a limitation of the sensitivity (or specificity) of the anti-K63 antibody and so we refrain from making strong conclusions on this point in the manuscript. In contrast, we believe the most significant observations in these experiments pertain to abundance of K48-linked polyubiquitin, which is appears much elevated in the presence of S57D and at the expense of K63-linked polymers. Additionally, we include new SILAC-MS analysis of polyubiquitin linkage abundance in these strains, and we have revised the text to include a more nuanced discussion of these observations – which includes consideration of the limitations associated with using ubiquitin replacement yeast strains, where ubiquitin is exclusively S57A or S57D. We hope reviewer #4 will agree that these revisions have helped to strengthen the manuscript, and that despite these limitations the results are still significant and worthy of publication.

Instead, the authors compare kinase knockout phenotypes to the previous Ser51Asp and Ser51Ala phenotypes. Unfortunately, even these are performed in a rather haphazard fashion. For example, although Ser57Asp was shown to produce a robust heat tolerance phenotype and also a tunicamycin sensitivity (Figure 1A-C and S1A), in Figure S3C wildtype cells were only compared in this regard to Δsks1Δvhs1 and not also to SksI/VhsI over-expression (or to Ppz1/2 knockout, the phosphatase identified by this group in Lee at al., (2017) to be responsible for Ser57P dephosphorylation and surprisingly not mentioned or utilised in this paper).

These are excellent points raised by reviewer #4. While our data do implicate Ser57 phosphorylation of ubiquitin in thermal tolerance, our analysis did not uncover a kinase responsible for this particular phenotype. We suspect that there may be additional, as yet unidentified ubiquitin kinases which may confer thermal tolerance.

Please note that we have also decided to remove our analysis of growth in the presence of tunicamycin. The tunicamycin resistance phenotype of *Δsks1Δvhs1* double mutant cells is subtle and will require additional genetic analysis in follow-up studies. However, we do not believe this exclusion changes the main conclusions or significance of the manuscript.

Examining a role for Ppz phosphatases as antagonists of these kinases is also an excellent idea. While we have some genetic data to support this relationship, we think this will require a deeper genetic and biochemical analysis that is outside the scope of the current manuscript.

[Editors' note: further revisions were suggested prior to acceptance, as described below.]

1) The new proteomics results, as disappointing as you might find them, are an important part of the story and should constitute part of a main figure and not be buried in a Supplementary file. They should be the basis of an important discussion point (see below).

We were advised by *eLife* editorial staff that tables cannot be incorporated into figures. As such, we decided to move the data from Supplementary file 3 into Table 1, which is now part of the main article. To link this to Figure 4, we now reference Table 1 in the Figure 4 legend (and vice versa). Additionally, to highlight this further, we have added the chromatography data that is the basis of quantification, which is now displayed in Figure 4H. We agree with reviewers that these revisions help to emphasize this important result.

At the request of reviewer 1, we have also log transformed the data in Table 1, to make it more consistent with our treatment of other similar data throughout the manuscript. This is described in the revised legend for Table 1.

2) Instead of a result and Discussion section, the paper should have a stand-alone Discussion section, where these concerns should be thoroughly put forward.

We agree, and we have implemented this suggestion in the revised manuscript.

3) In this discussion you should explicitly state that you are asking readers to accept the proposition that the phenotype in question arises from a small Sks1/Vhs1-dependent pool of Ub_pS57, which is undetectable on the background levels of Ub_pS57, which are themselves a very small fraction of Ub.

We have now expanded our discussion and included a more thorough explanation of our proposed interpretation, as well as the limitations associated with it. We have also included additional discussion of the limitations associated with genetic analysis of ubiquitin replacement yeast strains. These revisions are all found in the new Discussion section of the manuscript.